# Bringing Image Structure to Video via Frame-Clip Consistency of Object Tokens

**Elad Ben-Avraham**
Tel Aviv University
eladba4@gmail.com

**Roei Herzig**
Tel Aviv University, IBM Research
roeiherz@gmail.com

**Karttikeya Mangalam**
UC Berkeley
mangalam@cs.berkeley.edu

**Amir Bar**
Tel Aviv University
amirb4r@gmail.com

**Anna Rohrbach**
UC Berkeley
anna.rohrbach@berkeley.edu

**Leonid Karlinsky**
MIT-IBM Lab
leonidka@ibm.com

**Trevor Darrell**
UC Berkeley
trevordarrell@berkeley.edu

**Amir Globerson**
Tel Aviv University, Google Research
gamir@tauex.tau.ac.il

## Abstract

Recent action recognition models have achieved impressive results by integrating objects, their locations and interactions. However, obtaining dense structured annotations for each frame is tedious and time-consuming, making these methods expensive to train and less scalable. On the other hand, one does often have access to a small set of annotated images, either within or outside the domain of interest. Here we ask how such images can be leveraged for downstream video understanding tasks. We propose a learning framework StructureViT (SViT for short), which demonstrates how utilizing the structure of a small number of images only available during training can improve a video model. SViT relies on two key insights. First, as both images and videos contain structured information, we enrich a transformer model with a set of *object tokens* that can be used across images and videos. Second, the scene representations of individual frames in video should "align" with those of still images. This is achieved via a *Frame-Clip Consistency* loss, which ensures the flow of structured information between images and videos. We explore a particular instantiation of scene structure, namely a *Hand-Object Graph*, consisting of hands and objects with their locations as nodes, and physical relations of contact/no-contact as edges. SViT shows strong performance improvements on multiple video understanding tasks and datasets, including the first place in the Ego4D CVPR'22 Point of No Return Temporal Localization Challenge. For code and pretrained models, visit the project page at https://eladb3.github.io/SViT/.

## 1   Introduction

Semantic understanding of videos is a key challenge for machine vision and artificial intelligence. It is intuitive that video models should benefit from incorporating scene structure, e.g., the objects that appear in a video, their attributes, and the way they interact. Indeed, several works [5, 25, 26, 34, 35, 42, 81, 83, 89] have demonstrated that incorporating structured representations into models improves both performance and sample efficiency. In particular, [34, 81, 89] showed how to incorporate structured representations by utilizing static image object annotations in videos.

Recently, vision transformers (ViT) [20] have emerged as the leading model for many vision applications [4, 21, 12]. This raises the question: how can structured representations be leveraged for video tasks in a video transformer? Past works [4, 21] have proposed video transformer models

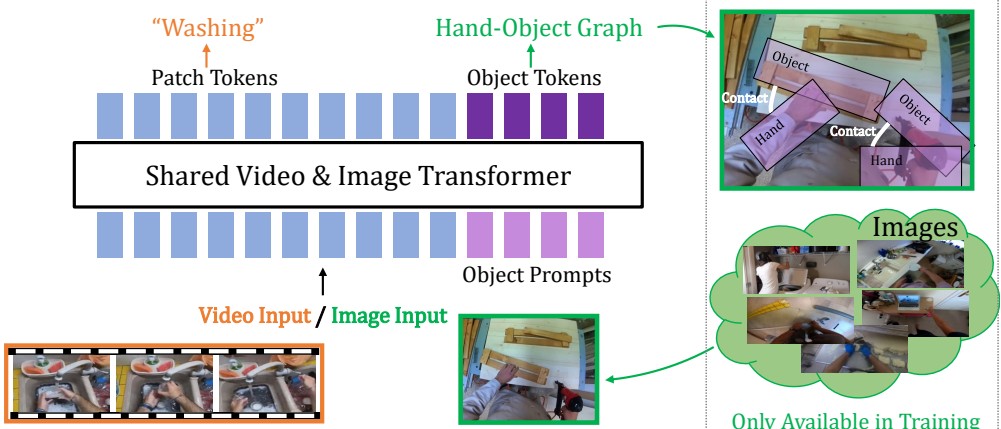

Figure 1: Our approach SViT brings structured scene representations from still images into video. We use the **HA**nd-**O**bject **G**raph (HAOG) annotations of still images that are only available during training, and videos, which may come from a different domain, with their downstream task annotations. We design a shared video & image transformer that can handle both video and image inputs. During training, given an image, the patch and object tokens are processed together, and the object prompts learn to predict the HAOG, whereas given a video input, the transformer predicts the video label (e.g. *Washing*) based on the patch tokens. During inference, the learned object prompts, which have captured the structured scene information from images, are used to predict the video label.

but these approaches were not focused on learning structured representations. A recent line of works [34, 81, 89] utilized objects for structured approaches. [83] developed an object-centric model based on region crop features from SlowFast [22], [34] proposed a new block that incorporates object representations, and [89] proposed a structured model for action detection. However, these approaches are limited because they use external pretrained detectors for the frames, which require additional processing time (also during inference). Moreover, they ignore the potential benefits of the connection between the detection and the other tasks, and it is unclear how to use them with more detailed structured annotations. Our approach, described next, builds on these prior works and proposes an effective way to *leverage scene structure in video transformers*, using a relatively *small* set of annotated static images.

Most structured transformer-based models for videos require structured annotations during training. For example, frame-level object bounding boxes are considered available in [5, 7, 34, 35, 59, 81]. When it comes to structured annotations of static images, there are numerous datasets with annotations such as boxes, visual relationships, attributes, and segmentations [10, 49, 59]. Since we cannot always expect to have annotated images that perfectly align with our downstream video task, how can these resources be used to build better transformer-based video models? In this work, we propose such an approach that is specifically designed for transformer architectures and offers an effective mechanism for improving video transformers by leveraging images with structured annotations that are *not necessarily associated* with the video domain and can be within or outside the domain of interest.

A natural first step towards image-video knowledge sharing is to use the same transformer model to process both: namely, to view images as "single frame videos", and feed them to a video transformer model. However, this still leaves two key questions: how to model structured information, and how to account for domain mismatch between images and videos. Towards this end, we introduce two key concepts: (i) A set of transformer "object-tokens" – additional tokens initialized from learned embeddings (we refer to these embeddings as "object prompts") that are meant to capture object-centric information in both still images and video. These tokens can be supervised based on the scene structure of images, and that information will also be propagated to videos. To formalize the image structure, we propose the HAnd-Object Graph (HAOG), a simple representation of the interactions between hands and objects in the scene. (ii) A novel "Frame-Clip Consistency loss" that ensures consistency between the "object-tokens" when they are part of a video vs. a still image. Moreover, as we show in our experiments, our approach does not require any alignment between object categories appearing in still images and the ones appearing in the videos of the target downstream task. We name our proposed approach StructureViT (*Bringing Scene **Structure** from Images to Video via Frame-Clip Consistency of Object **T**okens*, or SViT for short). See Figure 1 for an overview.

Prior work has explored several other video & image models. The I3D model [13] was one of the first successful attempts to leverage ImageNet architectures in video. They proposed to simply convert image 2D classification models into 3D ConvNets by inflating all the filters and pooling kernels.

Here, our focus is on learning the shared structured representations between the two domains in order to transfer this knowledge into videos. Several other methods have explored simultaneous image and video training in the context of multi-task [6] and multi-modal [27] learning. However, these works did not utilize structural information, nor did they promote consistency between image and video predictions – which, as we show, leads to significant gains in downstream video task performance.

To summarize, our main contributions are as follows: (i) we propose a new method for exploiting structured information present in images in order to improve performance of video understanding tasks; (ii) we propose a novel concept of "object tokens" to capture object-centric structure in transformer models of images and videos - a form of learned prompts to a video & image transformer, which during training are associated to the available structure representation of the still images. (iii) we introduce a new Frame-Clip Consistency loss, which promotes consistency between image-level and video-level predictions made by a shared transformer backbone. We show how this loss helps to drive performance improvements in downstream video tasks even in cases when image labels are unrelated to the video task; (iv) we demonstrate improved performance on several video understanding benchmarks, highlighting the effectiveness of the proposed approach.

## 2 The SViT Approach

Our SViT approach learns a structured representation that is exhibited in both static images and video. We consider the setting where the main goal is to learn video understanding downstream tasks while leveraging structured image data. In this paper, we focus on the following video tasks: action recognition, spatio-temporal action detection and object state classification & localization. In training, we assume that we have access to task-specific video labels as well as structured scene annotations for images. Based on the structured representations obtained by using the *object tokens* and regularized by the *Frame-Clip Consistency loss*, the inference is performed only for the downstream video task without explicitly using the structured image annotations.

We begin by describing an image annotation structure we refer to as Hand-Object Graphs (HAOG) (Section 2.1). We then introduce our SViT model (Section 2.2), and the Frame-Clip Consistency loss function (Section 2.3). Our method is schematically illustrated in Figure 2.

### 2.1 The Hand-Object Graph

As mentioned above, our motivation is to bring scene structure from still images into video. In order to achieve this, one component of this work is the construction of a semantic representation of the interactions between the hands and objects in the scene. Specifically, we propose to use a graph-structure we call *Hand-Object Graph* (HAOG), see Figure 1. The nodes of the HAOG represent two hands and two objects with their locations, whereas the edges correspond to physical properties such as contact/no contact. Formally, an HAOG is a tuple $(O, E)$ defined as follows:

**Nodes O:** A set $O$ of $n$ objects. Human-object interaction scenes, which we focus on in this work, usually consist of up to two hands and objects. Thus our node categories are "hand" and "object", and they may also contain attributes (e.g., spatial coordinates). We assume that each image comes with the bounding boxes describing the locations of the two hands, and the objects they interact with. The bounding boxes for the left and right hand are denoted as $b_1, b_2 \in \mathbb{R}^4$, respectively, while for the two corresponding objects interacting with the hands are denoted as $b_3, b_4$. We also assume access to four binary variables $e_1, \ldots, e_4 \in \{0, 1\}$ that indicate whether the corresponding hands or objects in fact exist in the image (e.g., if $e_1 = 0$ the left hand is not in the image, and $b_1$ should be ignored).

**Edges E:** The edges are represented as labeled edges between hand and object nodes. The edge labels describe the physical properties of the scene. In our current modeling, each edge is characterized by a physical property of "contact" or "no-contact". We denote two binary variables $c_1, c_2 \in \{0, 1\}$ that specify if each of the hands is in direct contact with the corresponding object (the left hand $b_1$ can only interact with the object $b_3$, similar for the right hand). In principle, we can also include properties such as distances between objects, 3D hand poses, etc., where available.

We note that images annotated consistent with our defined HAOG format can be found in several datasets [31, 59, 72], and we leverage these in our experiments.

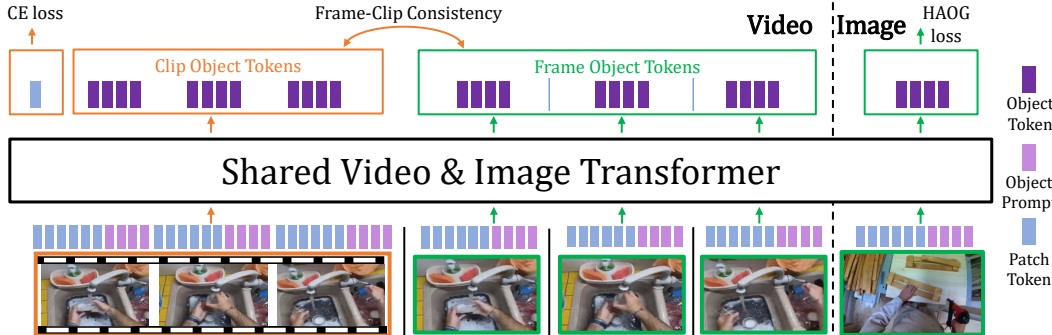

Figure 2: Our shared video & image transformer model processes two different types of tokens: standard patch tokens from the images and videos (**blue**) and the object prompts (**purple**), that are transformed into object tokens (**purple**) in the output. During training, the object tokens (**purple**) are trained to predict the HAOG for still images. For video frames that have no HAOG annotation, we use our "Frame-Clip" loss to ensure consistency between the "frame object tokens" (resulting from processing the frames separately) and the "clip object tokens" (resulting from processing the frame as part of the video). Last, the final video downstream task prediction results from applying a video downstream task head on the average of the patch tokens in the transformer output (after they have interacted with the clip object tokens (**purple**)).

## 2.2 The SViT Transformer Model

We next present the SViT model: a video transformer that employs *object tokens* as a means to capture structure in both images and videos. We begin by reviewing the video transformer architecture, which our model extends. Next, we explain how we process both images and videos. Finally, we describe object tokens and how they are used in videos and still images.

**Shared Video & Image Transformer**. Video transformers [4, 11, 21, 34] extend the Vision Transformer model to the temporal domain. Similar to vision transformers, the input is first "patchified" but with temporally extended 3-D convolution (instead of 2-D) producing a down-sampled tensor $X$ of size $T \times H \times W \times d$. We refer to this as "patch tokens". Then, spatio-temporal position embeddings are added for providing location and time information. Next, multiple stacked self-attention blocks are applied repeatedly on the down-sampled tensor $X$ to produce the final vector representation using mean pooling over the patch tokens.

In our approach, we want to be able to process batches of images or videos. A key desideratum in this context is to be able to input both videos and still-images into the same model. Towards this end, our first change is to rely only on 2-D convolutions in the first transformer step instead of 3-D. This allows single images to be used as inputs without padding, and does not hurt performance in practice.

**Object tokens**. Our goal is to learn a shared structured representation between the video and the image domains. We do this by adding transformer tokens to represent objects. These tokens are functionally similar to the patch tokens, with two exceptions. First, their initial embedding is learned. We refer to these embeddings as "object prompts". Second, they are used to predict the structured representations when those are available (here, we use the Hand-Object Graphs structure for still images). An object token is used to predict properties of the corresponding node in the HAOG, and the concatenation of two object tokens is used to predict edge properties.

Next, we describe the model more formally. Let $n$ be the maximum number of modeled objects per frame (or still image). We have $n$ tokens for each frame, and thus a total of $T \times n$ tokens. The token for object $i$ at frame $t$ is initialized with the vector $\boldsymbol{o}_i + \boldsymbol{r}_t$ where $\boldsymbol{o}_i \in \mathbb{R}^d$ is a learned object prompt and $\boldsymbol{r}_t \in \mathbb{R}^d$ is a learned temporal positional embedding (the same one used for initializing the patch tokens). With these new tokens, we have $T \times (H \times W + n)$ tokens (i.e, vectors in $\mathbb{R}^d$), and these together will go through the standard self-attention layers.

We denote the operator that outputs the final representation of the object tokens by $F_O$, where for a video $V$ we have $F_O(V) \in \mathbb{R}^{T \times n \times d}$ and for an image $I$ we have $F_O(I) \in \mathbb{R}^{n \times d}$. The $F_O(I)$ representation is used to predict structured representations for single images. We also include a loss that makes single frame representations align with those of clip representations (see Section 2.3). The operator that outputs the final representation vector used for the video downstream task is denoted by $F_{CLS}$, where for video $V$ we have $F_{CLS}(V) \in \mathbb{R}^d$.

Finally, our method can be used on top of the most common video transformers (MViT [21], TimeSformer [11], Mformer [66], Video Swin [56]). For our experiments, we use the MViTv2 [53] model because it performs well empirically.

## 2.3 Losses and Training

During training we have a set of images annotated with HAOGs and videos with downstream task labels. We use these as inputs to our model (Section 2.2) and optimize the losses described below.

**Video loss**. Each training video $V$ has a corresponding ground-truth category $Y \in \{1, \ldots, K\}$. As noted above, we have a vector representation for the entire video denoted by $F_{CLS}(V)$. Thus, we simply use a neural network to predict logits for $Y$. Specifically, our predicted logits are: $\hat{Y} = FC(F_{CLS}(V)) \in \mathbb{R}^K$, where $FC$ is a trainable fully connected network. We then consider a standard cross-entropy between the predicted logits $\hat{Y}$ and true labels $Y$.

$$\mathcal{L}_{Vid} := \text{CE}(\text{Softmax}(Y, \widehat{Y})) \, , \tag{1}$$

**HAOG loss**. The transformer outputs a set of $n$ object tokens for each image. Since we have supervision for these objects, we predict structured representations from the tokens, and introduce a loss to optimize these predictions. Recall our notation from Section 2.1. We let the transformer use $n = 4$ object tokens which correspond to the two hands (bounding boxes $\boldsymbol{b}_1, \boldsymbol{b}_2$) and two manipulated objects (bounding boxes $\boldsymbol{b}_3, \boldsymbol{b}_4$). The $j^{th}$ token is used to predict a corresponding bounding box $b_j$, as well as the existence variable $e_j$. Formally, we let:

$$\hat{\boldsymbol{b}}_j = \text{FC}_{bb}(F_O^j(I)) \, , \; \hat{e}_j = \text{FC}_e(F_O^j(I)) \, , \tag{2}$$

where $FC_{bb}, FC_e$ are fully connected networks with four and one output respectively, $F_O^j(I)$ is the $j^{th}$ object token, and $I$ is the input image. Formally, this defines the loss of the nodes as follows:

$$\mathcal{L}_{Nodes} = \sum_{i=1}^{4} \text{BCE}(\text{Sigmoid}(\hat{e}_i), e_i) + e_i \left( \text{GIoU}(\hat{\boldsymbol{b}}_i, \boldsymbol{b}_i) + L_1(\hat{\boldsymbol{b}}_i, \boldsymbol{b}_i) \right) \tag{3}$$

where L1 is the standard L1 distance, and GIoU is used as in [69]. Furthermore, we predict the contact variable from the concatenation of the corresponding hand and object tokens. Namely for $j \in \{1, 2\}$:

$$\hat{c}_j = \text{Softmax}(\text{FC}_c([F_O^j(I), F_O^{j+2}(I)])) \tag{4}$$

where $FC_c$ is a fully connected with one output. Thus, we define the loss of the edges as follows:

$$\mathcal{L}_{Edges} = \sum_{i=1}^{2} \text{CE}(\hat{c}_i, c_i) \tag{5}$$

The final prediction loss is the sum of the node and edge losses:

$$\mathcal{L}_{HAOG} = \mathcal{L}_{Nodes} + \mathcal{L}_{Edges} \tag{6}$$

**Frame-Clip Consistency Loss**. Since we have different losses for images and video, the model could find a way to minimize the image loss in a way that only applies to the images and does not transfer to video.[1] To avoid this, we need to make sure that the video representation contains the same type of information as the still images, and in particular the structured information carried by the object tokens. To achieve this, we explicitly enforce the object tokens to be consistent across still images (frames) and videos (clips) using a frame-clip consistency loss. To calculate the loss, we process each video twice: once as a clip, and once as a batch of $T$ frames. Namely, a clip $V$ consisting of $T$ frames $I_1, \ldots, I_T$, will be processed once as $F_O(V)$ and once as a list of images $(F_O(I_1), \ldots, F_O(I_T))$. This results in two groups of $T \times n$ object tokens, the first are the *clip object tokens*, denoted as, $\mathcal{O}_{clip} := F_O(V) \in \mathbb{R}^{T \times n \times d}$, the second are the *frame object tokens*, denoted as $\mathcal{O}_{frames} := [F_O(v_1), \ldots, F_O(v_T)] \in \mathbb{R}^{T \times n \times d}$. Since both groups originate from the same input, each clip object token has a single corresponding frame object token. Namely, there is an alignment

---

[1]For example, it could "overfit" to relying on the positional embedding of the first frame only when minimizing image-related losses.

between the clip object tokens and the frame object tokens. Following the intuition that each clip object token should contain the information from its corresponding frame object token, we minimize the $L1$ distance between each clip object token and frame object token. Namely:

$$\mathcal{L}_{Con} := L_1(\mathcal{O}_{\text{clip}}, \mathcal{O}_{\text{frames}}) \tag{7}$$

**Overall loss.** The total loss consists of $\mathcal{L}_{Con}$, $\mathcal{L}_{HAOG}$ and $\mathcal{L}_{Vid}$, where $\mathcal{L}_{Vid}$ is the loss of the main task. Each of the three terms in the loss is multiplied by a hyper-parameter ($\lambda_{Con}$, $\lambda_{HAOG}$, $\lambda_{Vid}$), and the total loss is the weighted combination of the three terms:

$$\mathcal{L}_{\text{Total}} := \lambda_{Con}\mathcal{L}_{Con} + \lambda_{HAOG}\mathcal{L}_{HAOG} + \lambda_{Vid}\mathcal{L}_{Vid} \tag{8}$$

## 3 Experiments and Results

We begin by describing the datasets, implementation details, and baselines and variants. We then evaluate our approach on several benchmarks. Specifically, we consider the following tasks: Compositional Action Recognition (Section 3.4), Object State Change Classification & Localization (Section 3.5), Action Recognition (Section 3.6), and Spatio-Temporal Action Detection (Section 3.7).

### 3.1 Datasets

We first describe the datasets used for the downstream video tasks, and those used as "auxiliary" datasets with annotated images. We use the following video datasets: **(1) Something-Something v2 (SSv2)** [30] is a dataset containing 174 action categories of common human-object interactions. **(2) SomethingElse** [59] which exploits the compositional structure of SSv2, where a combination of a verb and a noun defines an action. We follow the official compositional split from [59], which assumes the set of noun-verb pairs available for training is disjoint from the set given at test time. **(3) Ego4D** [31] is a new large-scale dataset of more than 3,670 hours of video data, capturing the daily-life scenarios of more than 900 unique individuals from nine different countries around the world. **(4) Diving48** [54] contains 48 fine-grained categories of diving activities. **(5) Atomic Visual Actions (AVA)** [32] is a benchmark for human action detection, we report Mean Average Precision (mAP) on AVA-V2.2. For "auxiliary" image datasets we use the **Ego4D** [31], and the **100 Days of Hands (100DOH)** [72] datasets. We collected 79,921 annotated images from 100DOH, and 57,213 annotated images from Ego4D.[2] The AVA or SSv2 datasets are also used as "auxiliary" images in some experiments. For more details, see Section D.1 in the supplementary. For the image datasets, the image annotations are based on frames selected from videos, but these are often sparsely selected from the videos, so it is natural to treat them as annotated still images.[3]

### 3.2 Implementation Details

SViT is implemented in PyTorch, and the code is available on our project page. Our training recipes and code are based on the MViTv2 model, and were taken from `https://github.com/facebookresearch/mvit`. We pretrain the SViT model on the K400 [45] *video* dataset with our *auxiliary image* datasets. Then, we finetune on the target video understanding task (detailed in Section 3.1) together with the *auxiliary image* datasets and the SViT loss. Each training batch contains 64 images and 64 videos in order to minimize the overall loss in Equation 8. For more details about datasets and evaluation see Section C of Supplementary.

### 3.3 Baselines and SViT variants

**Baselines**. In the experiments, we compare SViT to several models from prior work reported for the corresponding datasets. These include the following approaches: *I3D* [13], *SlowFast* [22], and the state-of-the-art transformers – *Mformer* [66] and *MViTv2* [53]. In addition, we provide the *MViTv2 multi-task (MViTv2 MT)* baseline for comparing a naive multi-tasking approach to our model. This is a version of MViTv2 which performs video classification and HAOG prediction from images. The

---

[2]Comparatively, SomthingElse contains 2,550,700 frames, showing that image datasets are relatively small.
[3]The SomethingElse/SSv2 datasets has annotations for all clip frames, but we show in the experiments that SViT only needs a small percentage of these.

| Model | Boxes Annotations | Compositional Top-1 | Top-5 | Base Top-1 | Top-5 | Few-Shot 5-Shot | 10-Shot |
|---|---|---|---|---|---|---|---|
| SlowFast [22] | ✗ | 45.2 | 73.4 | 76.1 | 93.4 | 22.4 | 29.2 |
| TimeSformer [11] | ✗ | 44.2 | 76.8 | 79.5 | 95.6 | 24.6 | 33.8 |
| Mformer [66] | ✗ | 60.2 | 85.8 | 82.8 | 96.2 | 28.9 | 33.8 |
| MViTv2 [53] | ✗ | 63.3 | 87.5 | 83.7 | 96.8 | 32.7 | 40.2 |
| MViTv2 MT | ✓ | 63.8 | 87.3 | 85.2 | 97.1 | 33.7 | 41.4 |
| SViT-SFT | ✓ | 64.2 | 87.9 | 85.0 | 97.1 | 33.8 | 41.7 |
| SViT-DD | ✓ | 65.1 | 88.0 | 85.8 | 97.4 | 34.6 | 42.5 |
| SViT ×2% | ✓ | 65.6 | 88.1 | 85.9 | 97.4 | 34.4 | 43.1 |
| SViT-ID | ✓ | 65.8 | 88.3 | 85.6 | 97.3 | 34.4 | 42.6 |

Table 1: **Compositional and Few-Shot Action Recognition** on the "SomethingElse" dataset.

MViTv2 model is enhanced with two heads: the first head is composed of an MLP that predicts the video labels given a final representation vector, the second head consists of MLPs that predict the HAOG information (i.e., hands and object boxes, and contact information).

**SViT Variants**. As mentioned above, we use several sources of auxiliary image datasets. In some cases, we have auxiliary images that are frames from video understanding datasets. For example, in Ego4D, we have both videos and frames annotated from some of these videos. In this sense, the auxiliary data is "in-domain" for some of the video tasks. In order to evaluate the importance of this effect, we explore "Different Domain" (*SViT-DD*) training, which avoids using images from a given video task as auxiliary during finetuning and pretraining. Similarly, "In Domain" (*SViT-ID*) refers to the case where we use only images from the video task during finetuning, but not necessarily during pretraining . We consider another variant that does not use the auxiliary images during finetuning at all (but does use them during pre-training), and refer to it as *SViT-SFT*. Finally, to explore the effect of data-size in the "In Domain" setting, we evaluate a version *SViT ×X%* that only uses X% of the in-domain finetuning data. For MViTv2 MT, the auxiliary images correspond to the "In Domain" setting (i.e., same setting as for SViT-ID).

**Auxiliary Images datasets**. For SViT-DD, we generally use 100DOH and Ego4D for pretraining and finetuning, while for the Ego4D tasks, we only use 100DOH. SViT-ID uses 100DOH and Ego4D for pretraining and in-domain downstream frames for finetuning. The SViT-SFT pretraining setting is similar to that of SViT-DD. SViT ×X% setting is similar to that of SViT-ID. For more details, see Section D.1 in the supplementary.

### 3.4 Compositional & Few-Shot Action Recognition

Several video datasets define an action as a combination of a verb and a noun. One challenge then is to recognize combinations that were not seen during training. This "compositional" setting was explored in the "SomethingElse" dataset [59], where verb-noun combinations in the test data do not occur in the training data. The split contains 174 classes with 54,919/54,876 videos for training/validation. We also evaluate on the few-shot compositional action recognition task in [59] (See Section C.3 in supplementary).

Table 1 reports the results for these two tasks. All of our variants outperform all prior works and the MViTv2 MT on the *Compositional* and *Few-shot* task. From the table, we can see that using 2% of the in-domain data as auxiliary images (SViT×2%), is almost comparable to using all the available in-domain images (SViT-ID). Furthermore, the results are comparable to SViT-DD (0.7 difference) when trained on auxiliary images from different domains. This suggests that our approach optimizes the use of structural information across domains. Last, SViT-DD outperforms SViT-SFT when using the same data, demonstrating the value of finetuning video and image together.

### 3.5 Object State Change Classification and Localization

Hands and objects are key elements in human activity. Two tasks related to hand-object interaction have recently been introduced in the Ego4D [31] dataset. The first is temporal localization, which is defined as finding key frames in a video clip that indicate a change in object state. The second is object state change classification, which indicates whether an object state has changed or not. SViT won the first place at the Ego4D CVPR'22 Point of No Return Temporal Localization Challenge.

(a) Object State Change in Ego4D.

| Model | Temporal loc. error | Cls. top-1 |
|---|---|---|
| Bi-LSTM [†] | 0.790 | 65.3 |
| BMN[†] | 0.780 | - |
| I3D ResNet-50[†] | 0.739 | 68.7 |
| MViT-v2[†] | 0.702 | 71.6 |
| MViTv2 MT | 0.678 | 71.1 |
| SViT-SFT | 0.654 (−.048) | 70.4 (−1.2) |
| SViT-DD | 0.646 (−.056) | 73.6 (+2) |
| SViT ×2% | 0.649 (−.053) | 73.4 (+1.8) |
| SViT-ID | 0.64 (−.062) | 73.8 (+2.2) |

(b) Action Recognition in SSv2.

| Model | Top-1 |
|---|---|
| SlowFast, R101[†] | 63.1 |
| ViViT-L [†] | 65.4 |
| MViTv1 [†] | 64.7 |
| MViTv2 [†] | 68.2 |
| MViTv2 MT | 68.3 |
| SViT-SFT | 68.9 (+0.7) |
| SViT-DD | 69.2 (+1) |
| SViT ×2% | 69.4 (+1.2) |
| SViT-ID | 69.7 (+1.5) |

(c) Action Recognition in Diving48.

| Model | Top-1 |
|---|---|
| SlowFast, R101 [†] | 77.6 |
| TimeSformer [†] | 74.9 |
| TimeSformer-HR [†] | 78.0 |
| MViTv2 [†] | 73.1 |
| MViTv2 MT | 75.5 |
| SViT-SFT | 77.9 (+4.8) |
| SViT-DD | 79.8 (+6.7) |

Table 2: **Results on Ego4D, SSv2, and Diving48 datasets.** Evaluation metrics for the Ego4D object state change task are absolute error in seconds (temporal loc. error. Lower is better) for temporal localization, and top-1 accuracy (Cls. top-1) for state change classification. We denote methods that do not use any additional structured annotation with [†].

Table 2a reports results on the temporal localization and object state change classification tasks on the Ego4D dataset. We observe that SViT-ID, SViT×2% and SViT-DD perform better than SViT-SFT and are very close. These results are consistent with those presented in Section 3.4, which indicates that our method successfully leverages structural information across domains, and that a joint image and video finetuning is beneficial.

## 3.6 Action Recognition

Table 2b and Table 2c report the results for standard action recognition task on the SSv2 and Diving48 datasets respectively. It can be seen that in Diving48 dataset, our method improves over the MViT-v2 baseline by 6.7%, outperforming all the methods. This demonstrates that our method has some ability to generalize the learned structural information of hand-object interactions even to fine-grained "human-motion" tasks such as Diving48. We also note that Diving48 has no structured annotations available, thus SViT-ID (and SViT×2%) cannot be provided. On SSv2, SViT-ID improves the baseline by 1.5%, SViT×2% improves by 1.2%, and SViT-DD improves by 1.0%. As we observed significant improvements on the more challenging SomethingElse setup, we hypothesize that SomethingElse is more likely to benefit from the ability to generalize to unseen combinations of objects and hands exhibited by SViT. We provide visualizations of the HAOGs predicted by SViT in Section B of the supplementary.

## 3.7 Spatio-temporal Action Detection

Table 3 reports the results for spatio-temporal action detection on the AVA dataset. In the literature, the action detection task on AVA is formulated as a two stage prediction procedure. The first step is the detection of bounding boxes, which are obtained through an off-the-shelf pretrained person detector. The second step involves predicting the action being performed at each detected bounding box. The performance is benchmarked on the end result of these steps and is measured with the Mean Average Precision (MAP) metric. Typically, for fair comparison, the detected person boxes are kept identical across approaches and hence the final performance depends directly on the ability of the approach to utilize the video and box information. We observe that all SViT variants improve over the MViT-v2 baseline. Specifically, SViT-ID improves the MViT-v2 baseline by 1.7, SViT-DD improves by 1.5, and SViT-SFT improves by 1.0.

| Model | mAP |
|---|---|
| SlowFast [22] [†] | 23.8 |
| MViTv2 [†] | 26.8 |
| MViTv2 MT | 27.2 |
| SViT-SFT | 27.8 (+1.0) |
| SViT-DD | 28.3 (+1.5) |
| SViT-ID | 28.5 (+1.7) |

Table 3: **Spatio-temporal Action Detection** on AVA-V2.2.

## 3.8 Ablations

We perform a comprehensive ablation study on the compositional action recognition task [59] on the SomethingElse dataset to measure the contribution of the different SViT components (Table 4).

| (a) Method components | | (b) Images Amount | | (c) Absence of hands/objects | | | |
|---|---|---|---|---|---|---|---|
| **Model** | **Top-1** | **% of data** | **Top-1** | **Model** | **Objects** | **Hands** | **Top-1** |
| MViTv2 | 63.3 | 2 | 65.6 | MViTv2 | ✓ | ✗ | 64.6 |
| MViTv2 MT | 63.8 | 25 | 65.6 | SViT-ID | ✓ | ✗ | 66.5 |
| MViTv2+Object Tokens (OT) | 65.0 | 50 | 65.7 | MViTv2 | ✗ | ✓ | 64.9 |
| MViTv2 +OT+FC Loss (SViT) | 65.8 | 100 | 65.8 | SViT-ID | ✗ | ✓ | 67.0 |

Table 4: **Ablations.** We show (a) Contributions of SViT components. (b) Amount of annotated images used in training. (c) Examine the robustness of SViT to clips that does not contain hands/contains only hands. The experiments are performed on the SomethingElse split. For more ablations, see A in supplementary.

**Components of the SViT model**. In Table 4a, we ablate our model to show the significance of the individual components. First, we train a standard MViTv2 in a multi-task setting (MViTv2 MT), where we use videos for video action recognition, and images for HAOG prediction. There is a small improvement (+0.5) over the baseline. We then take the MViTv2 and add the object tokens and corresponding SViT loss; this leads to a more significant improvement (+1.7). Last, when adding both the consistency loss and the object tokens, we observe the largest improvement (+2.5).

**The amount of in-domain data**. In Table 4b, we demonstrate the effect of the amount of in-domain image data used during training. Specifically, since "SomethingElse" includes HAOG annotations for each frame, we train our method with different amount of images (2%, 25%, 50%, and 100%) from the "SomthingElse" dataset. It can be seen that our approach takes advantage of the structured information within a small amount of data and, as a result, has enhanced sample-efficiency.

**Robustness to absence of hands/objects**. In table Table 4c we verify whether our approach works in the cases where no hands or objects are visible, we conducted experiments on SomethingElse (ground-truth annotations available): (i) After filtering the videos containing (annotated) hands, we tested our model and the MViTv2 model on the filtered split. MViTv2 achieved an accuracy of 64.6 while our model achieved 66.5. This implies an improvement of +1.9. (ii) After filtering the videos containing (annotated) objects in more than 40% of the frames, we tested our model and the MViTv2 model on the filtered split. MViTv2 achieved an accuracy of 64.9 while our model achieved 67.0. This implies an improvement of +2.1. We can observe that our model outperforms the baseline even when there are no objects or hands in the videos, demonstrating the robustness of our approach. The total improvement (+1.9 and +2.1) is also a little bit lower than before the filtering (+2.5), which indicates that there has been a slight degradation. Nevertheless, hand-object graphs remain valuable.

**How important are the HAOGs**. To examine how important is the information provided by the HAOG, we suggest learning HAOGs without any useful information. Thus, we run an experiment in which the HAOGs are completely random. This means that, for each image, a random HAOG is generated by sampling the boxes and their relationships uniformly. We refer to this experiment as SViT-Random-ID. On the SomethingElse dataset, the SViT-Random-ID result is 50.6 while the SViT-ID is 65.8. This shows that predicting actual HAOG attributes provides important information that can be helpful for video downstream tasks.

## 4 Related Work

**Joint Training with Images and Videos**. Vision Transformers [20] are versatile architectures designed to be flexible in terms of input sequence length. Aside from the data "'patchification'" layer, the rest of the transformer is agnostic to the domain of the input. Consequently, an exciting direction of joint training image-video has emerged in computer vision, a direction previously less natural with convolutional architectures. Specifically, this allows for training networks simultaneously on image and video, with most of the network parameters shared across both domains. Recently, multi-modal models became widely popular, from shared backbones [15, 57, 61] to separate encoders for each input modality [74, 77], and even non-vision modalities [39, 40]. In our case, we focus on video and image input, similar to [6, 27, 87]. In contrast to these works, our training approach utilizes two domains, while one of them (image) is used only during training and is supervised by a different task. Additionally, as we demonstrate here, we are not greatly impacted by domain mismatches between images and videos, as in previous works. In a related area of research, multi-task learning [14]

involves developing models that predict output for multiple tasks using a common input. Our work is different since it uses two inputs from different domains, images and videos. Similarly, our work can be viewed as a form of auxiliary task, as already shown in many works [29, 38, 73]. However, it differs in that the shared object tokens are utilized to learn the shared structured representation between video and image domains, which enhances the self-attention layers with the structured representation for the main video task.

**Learning from video using other modalities**. Researchers have proposed different approaches to learn from videos using audio and natural language [1, 3, 19, 62, 64, 65, 68, 82]. Vision tasks like Video Captioning [19, 24, 65, 82] and Visual Quesion Answering [2, 79] require understanding of both visual and natural language. Another line of vision and language works explored object labels, supervision of textual descriptions, or the use of region crops [50, 52], in contrast to SViT, which does not require any of these components. Additionally, since high quality captions are expensive, researchers have proposed self-supervised ways to learn from both modalities without labeled data [62, 61, 68]. In these previous works, the two modalities must be consistent and aligned (e.g., text, audio, and RGB should be consistent for same sample). However, in our work, we do not require that the annotated images be precisely aligned with our downstream video task. Last, other new works [16, 70] explored different structured representations for videos, different from what we do, which is to leverage structure from images to videos.

**Structured Models**. Structured models have recently been successfully applied in many computer vision applications: video relation detection [55, 71, 76], vision and language [17, 51, 52, 78], relational reasoning [8, 9, 36, 48, 41, 67, 85, 86], human-object interactions [23, 44, 84], and even image & video generation [7, 33, 43]. A recent line of works [5, 25, 26, 34, 35, 42, 58, 60, 63, 75, 80, 81, 83, 89] focuses on capturing spatio-temporal interactions for video understanding. Some of these works [34, 81, 83, 89] used static image object annotations to build the representations. However, these works differ from ours. [83] proposed the Object Transformers (OT) for long form video understanding, which model a video as a set of objects by extracting region crops from SlowFast [22] based on bounding boxes from an external detector. Contrary to this work, our transformer contains both patch tokens and object tokens, whereas the OT contains only object tokens. Additionally, our model learns the object-centric representation without the need for an external detector. Another work [89], proposed a structured model for action detection that explicitly utilizes image annotations. They use a GCN [47] to reason about actors and objects interaction, on top of an I3D network. Unlike this work, we examine how image annotations can be exploited as part of a video transformer model.

**Video Understanding Models**. We work on several well-benchmarked down-stream video tasks such as compositional action recognition on Something-Else dataset [59] and action recognition with focus on temporal cues de-emphasizing appearance such as in Something-Something V2 dataset [30] and Diving48 [54]. Several recent video transfromer works performed well on SSv2, such as ViViT [4], MViT [21], MFormer [66], TimeSformer [11], MViTv2 [53], and Video Swin [56]. We choose to work with MViTv2, although our method can be used on top of any of these. Additionally, we also adapt our proposed method to interesting new tasks on the Ego4D dataset [31]. We believe such fine-grained video understanding tasks require richer scene and object semantic reasoning, a requirement well afforded by our proposed Hand-Object Graph.

## 5   Conclusion

Video understanding is a key element of human visual perception, but modeling remains a challenge for machine vision. In this work, we demonstrated the importance of learning from the scene structure of a small set of images to facilitate video learning within or outside the domain of interest. According to our empirical study, our SViT approach improves performance on four video understanding tasks. Note, that we did not prioritize making the HAOG annotations richer, i.e., it may be possible to add other physical properties to improve structure modeling. In addition, we noticed that leveraging the structure is effective even with $2\%$ of the in-domain frames, as well as when applying it from outside the domain. We leave for future work the challenge of utilizing different physical properties and more complex structures. Regarding potential impact of the method, We do not anticipate a specific negative impact, but, as with any Machine Learning method, we recommend to exercise caution.

**Acknowledgements**

This project has received funding from the European Research Council (ERC) under the European Unions Horizon 2020 research and innovation programme (grant ERC HOLI 819080). Prof. Darrell's group was supported in part by DoD including DARPA's Semafor, PTG and/or LwLL programs, as well as BAIR's industrial alliance programs.

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
