# Supplementary Material for "Bringing Image Scene Structure to Video via Frame-Clip Consistency of Object Tokens"

In this supplementary file, we provide additional information about our experimental results, qualitative examples, implementation details, and datasets. Specifically, Section A provides more experiment results, Section B provides qualitative visualizations to illustrate our approach, Section C provides additional implementation details, and Section D provides additional datasets details.

## A  Additional Experiment Results

First, we discuss the pretraining and finetuning of the SViT model variants (Section A.1). Next, we present additional ablations (Section A.2) we performed in order to test the contribution of the different SViT components.

### A.1  SViT Variants

As explained in Section 3.3, we use several SViT variants, see Table 5 for details. Additionally, Table 7 provides a detailed listing of the datasets we used for each SViT variant and task.

| Variant | Pretrain Auxiliary Image Data | Finetune Auxiliary Image Data |
|---------|-------------------------------|-------------------------------|
| SViT-ID | In-Domain + Different-Domain | In-Domain |
| SViT-DD | Different-Domain | Different-Domain |
| SViT-SFT | Different-Domain | None |

Table 5: **SViT Variants**

### A.2  Additional Experiments

Next, we provide additional experiments in Table 6.

**Pretraining and finetuning using in-domain auxiliary images**. As part of finetuning, SViT-ID makes use of in-domain auxiliary image data, whereas for pretraining, it uses a different domain and in-domain of image data (see Table 5). It is interesting to see what happens if one uses only in-domain data for both pretraining and finetuning, and no data from external datasets. Towards this end, we consider a SViT training setup where only the in-domain images are used for both pre-training and fine-tuning. We do this both for SmthElse and Ego4D. Results are shown in Tables 6a and 6b. It can be seen that results with only in-domain image data (both pretraining and finetuning) are quite close to those of using both different and in-domain data. Thus, we conclude that the performance improvement of the in-domain and different domain are similar, demonstrating that our method is capable of benefiting both from in-domain and different-domain data.

**Pertraining and Finetuning with or without auxiliary images**. In Table 6d, we check the effect of using the auxiliary image data during the pretraining and finetuning for the standard SViT-ID model. Namely, during the pretraining and fine tuning processes, we examine each combination of using or not using auxiliary image data. We can see that doing finetuning and pretraining with auxiliary images yields the best results (65.8). Alternatively, only pretraining with auxiliary images (64.2) or only finetuning with auxiliary images (64.4) are advantageous compared to not using any auxiliary images. In fact, not using images at all is equivalent to MViT-v2 with additional tokens, and has no benefit over MViT-v2.

**Consistency Loss**. In Table 6f we consider two different types of frame-clip consistencies. The first is patch token consistency, which is simply replacing the object tokens with the patch tokens in the frame-clip consistency loss and the second is object token consistency (as described in Equation 7 in the main paper). Comparing the results to not using any frame-clip consistency (65.0), we note that object token consistency is beneficial (+0.8), but patch consistency decreases the performance (-6.3). We hypothesize that patch consistency may reduce patch diversification within the frames, thus reducing performance. A recent study [28] has demonstrated that vision transformers tend

(a) ID pretraining on **SomethingElse**

| Model | Pretraining Aux. Images | Top-1 |
|---|---|---|
| SViT-ID | diff-domain | 65.8 |
| SViT $\times 2\%$ | diff-domain | 65.1 |
| SViT-ID | in-domain | 65.6 |
| SViT $\times 2\%$ | in-domain | 65.1 |

(b) ID pretraining on **Ego4D**

| Model | Pretraining Aux. Images | Temporal Loc. Error | Cls. Top-1 |
|---|---|---|---|
| SViT-ID | diff-domain | 0.64 | 73.8 |
| SViT-ID | in-domain | 0.642 | 71.9 |

(c) HAOG Attributes

| Model | Top-1 |
|---|---|
| SViT -Contact | 65.5 |
| SViT -Contact -Corresp. | 65.3 |
| SViT +Geometry | 65.7 |
| SViT | 65.8 |

(d) Auxiliary images during pretraining and finetuning

| Model | Pretrain | Finetune | Top-1 |
|---|---|---|---|
| SViT-ID | w/o Images | w/o Images | 63.4 |
| SViT-ID | w/o Images | w/ Images | 64.4 |
| SViT-ID | w/ Images | w/o Images | 64.2 |
| SViT-ID | w/ Images | w/ Images | 65.8 |

(e) Images-Videos Ratio

| Images/Videos | Top-1 |
|---|---|
| 1/1 | 65.6 |
| 1/10 | 64.8 |
| 1/100 | 61.6 |

(f) Consistency loss

| Model | Frame-Clip Consistency Type | Top-1 |
|---|---|---|
| SViT-ID | Patches | 58.7 |
| SViT-ID | Objects | 65.8 |
| SViT-ID | None | 65.0 |

Table 6: **Additional Experiments.** Unless otherwise noted, all experiments were performed on the "SomethingElse" dataset. In tables 6a and 6b, "diff-domain" refers to different-domain.

to map different patches into similar latent representations, which results in information loss and performance degradation.

**The minimum requirements to make the model work**. In Table 6e we aim to systematically demonstrate the minimum requirements necessary for the model to function reasonably well. We explored the minimum requirements to make our model work. We provide the following experiments: (i) A ratio of 1 image to X videos (where X is 1,10, and 100): 65.6 (1-to-1), 64.8 (1-to-10), and 61.55 (1-to-100). The results indicate that using too few annotated images (1 image per 100 videos) may result in degradation, but using a relatively small number of images (1 image per 10 videos) is sufficient to achieve good improvement.

**HAOG attributes**. We use the object tokens to predict several aspects of the HAOG. In Table 6c we consider several variations on this prediction. The model "SViT-Contact" does not predict the contact information, resulting in $-0.3\%$ drop compared to SViT. The model "SViT+Geometry" adds to SViT additional heads that predict distances between all HAOG objects (to explicitly add geometric bias). This does not affect performance. Finally, we explore what happens when we do not provide the information about the identity of bounding boxes in the training data. Namely, we treat them as four boxes, and ask the model to match those to the object tokens, via a matching losss as in [12]. This is model "SViT-Contact -Corresp." which is quite close to "SViT-Contact", indicating that SViT can perhaps be trained with weaker labels.

**Frame-clip consistency loss**. In order to validate frame-clip consistency loss, we evaluate the following experiments: (i) We perturb the image temporal position without consistency loss. We refer to this version as SViT-Perturb (and similarly SViT-Perturb-DD and SViT-Perturb-ID). (ii) The HAOG annotations are extrapolated from one random frame of a video. This serves as additional supervision (without the consistency loss) since the HAOG annotations correspond to the video frames (we note that SViT does not require such correspondence since it uses only HAOG annotations from single images). (iii) We predict the HAOG of a random frame in a video, and then duplicate it over the

temporal dimension and use it in the same manner as in the consistency loss. We find that these three baselines lead to worse performance. For the first experiment, we obtained SViT-Perturb-DD with 64.1 (while SViT-DD obtained 65.1), and SViT-Perturb-ID with 65.2 (while SViT-ID with consistency loss got 65.8). For the second experiment, the proposed baseline achieved 65.0 compared to our SViT-ID, which achieved 65.8 (and does not require correspondence). Taken together, this demonstrates the importance of our frame-clip consistency loss.

**HAOG "distillation"**. In this experiment we first train a vision transformer to predict the HAOG task alone and then fine-tune its backbone network for the video-related task. We evaluated and achieved 62.3 compared to 63.3 of MViTv2. This indicates that training HAOGs as a form of "distillation", as we do in SViT, is indeed important.

**The object tokens representations**. To verify what the object tokens learned, we can evaluate the ability of object tokens to be utilized explicitly for the auxiliary task as a simple detector of hands and objects in images. This is accomplished by predicting the detections on SomethingElse based on the learned object tokens. The learned object tokens were compared to the MViTv2 model extended with regression and detection heads. Our model achieved an mAP of 16.8, while the proposed baseline achieved a similar result with an mAP of 15.5. These results suggest that the object tokens learn meaningful and useful representations.

**Computational cost**. The cost computation of our approach is relatively small compared to MViTv2: +0.4% in Giga ($10^9$) FLOPs (MViTv2 with 64.5, while SViT with 64.7) and +5% in Mega ($10^6$) parameters (MViTv2 with 34.4, while SViT with 36.1).

**Hands in Diving48**. To further analyze SViT performance on Diving48 we provide an experiment in which we remove the "hand" tokens from the HAOG annotation during training, and an experiment in which we remove the "object" tokens from the HAOG annotation during training. The result for removing the "hand" tokens is a degradation of 1.7 in top-1 accuracy, and for removing the "object" tokens is 0.6. We hypothesize that the "human" boxes provide a prior for localization that helps in classifying diving actions (diving actions could be classified based on appearance and pose).

## B   Qualitative Visualizations

We present qualitative visualizations of the SViT-ID "object tokens" in videos for the following datasets: SomethingElse, Ego4D, and Diving48 in Figure 3. As explained in the main paper, two tokens correspond to the "right hand" and "left hand", whereas the other two tokens correspond to the objects they are interacting with. We can see that the "object tokens" are used to detect relevant objects, even on Diving48, where there is no human-object interaction. In Diving48, we observed a few interesting phenomena: (i) the diving persons are recognized as hands. This may be explain by the fact that skin has shared visual features with hands. (ii) The objects recognized in low scores indicate that the scene does not contain any objects. Overall, despite the issues we raised above, our model performed well on Diving48. This is evidence that our method is robust to multiple domains that do not necessarily fit with human-object graph structure (HAOG), and that it still benefits from these learned object tokens through the attention.

## C   Additional Implementation Details

Our SViT model can be used on top of the most common video transformers (MViT [21], TimeS-former [11], Mformer [66], Video Swin [56]). For our experiments, we choose the MViTv2 [53] model because it performs well empirically. These are all implemented based on the MViTv2 [53] library (available at `https://github.com/facebookresearch/mvit`), and we implement SViT based on this repository. Additionally, in $\mathcal{L}_{Nodes}$ (see 3) we include weights for the $L_1$, $BCE$ and $GIoU$ losses, set to 5, 1 and 2 respectively (across all datasets). Next, we elaborate on the additional implementation details for each dataset, including details about the dataset description, optimization, and training and inference.

### C.1   Ego4D

**Dataset**. Ego4D [31] is a new large-scale dataset of more than 3,670 hours of video data, capturing the daily-life scenarios of more than 900 unique individuals from nine different countries around the

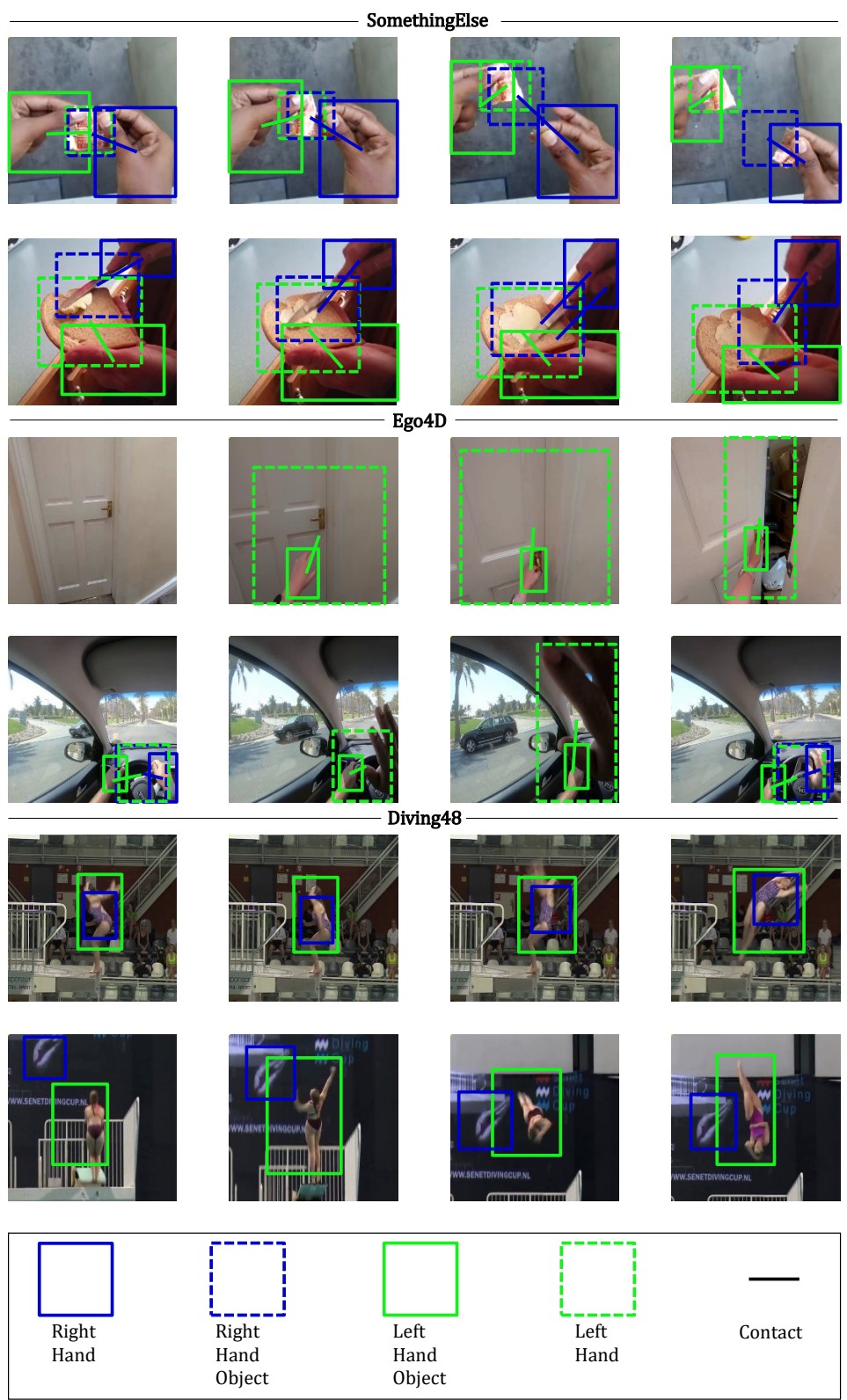

Figure 3: **Qualitative visualization of the "object tokens"**. The left object and hand prediction is visualized with **green bounding box**, the right object and hand prediction is visualized with **blue bounding box**. A line connecting between two boxes indicates a "contact" prediction between the two objects. Only boxes with a score greater than 0.6 are plotted.

world. The videos contain audio, 3D meshes of the environment, eye gaze, stereo and/or synchronized videos from multiple egocentric cameras.

**Metrics**. In the Object State Change Temporal Localization task, the absolute error (in seconds) is used for evaluation. In the Object State Change Classification task, the top-1 accuracy is used for evaluation.

**Optimization details**. We train using 16 frames with sample rate 4 and batch-size 128 (comprising 64 videos and 64 images) on 8 RTX 3090 GPUs. We train our network for 10 epochs with Adam optimizer [46] with a momentum of $9e-1$ and Gamma $1e-1$. Following [53], we use $lr = 1e-4$ with half-period cosine decay. Additionally, we used Automatic Mixed Precision, which is implemented by PyTorch.

**Training details**. We use a standard crop size of 224, and we jitter the scales from 256 to 320. Additionally, we set $\lambda_{Con} = 10, \lambda_{HAOG} = 5, \lambda_{Vid} = 1$.

**Inference details**. We follow the official evaluation, both for the state change temporal localization and the state change classification tasks, available at https://github.com/EGO4D/hands-and-objects.

## C.2 Diving48

**Dataset**. Diving48 [54] contains 16K training and 3K testing videos spanning 48 fine-grained diving categories of diving activities. For all of these datasets, we use standard classification accuracy as our main performance metric.

**Optimization details**. We train using 16 frames with sample rate 4 and batch-size 128 (comprising 64 videos and 64 images) on 8 RTX 3090 GPUs. We train our network for 10 epochs with Adam optimizer [46] with a momentum of $9e - 1$ and Gamma $1e - 1$. Following [53], we use $lr = 1e - 4$ with half-period cosine decay.

**Training details**. We use a standard crop size of 224 for the standard model and jitter the scales from 256 to 320. Additionally, we use RandomFlip augmentation. Finally, we sample the $T$ frames from the start and end diving annotation time, following [88]. We set $\lambda_{Con} = 10, \lambda_{HAOG} = 5, \lambda_{Vid} = 1$.

**Inference details**. We take 3 spatial crops per single clip to form predictions over a single video in testing, as in [11].

## C.3 SomethingElse

**Dataset**. The SomethingElse dataset [59] contains 174 action categories with 54,919 training and 57,876 validation samples. The proposed compositional [59] split in this dataset provides disjoint combinations of a verb (action) and noun (object) in the training and testing sets. This split defines two disjoint groups of nouns $\{\mathcal{A}, \mathcal{B}\}$ and verbs $\{1, 2\}$. Given the splits of groups, they combine the training set as $1\mathcal{A} + 2\mathcal{B}$, while the validation set is constructed by flipping the combination into $1\mathcal{B} + 2\mathcal{A}$. In this way, different combinations of verbs and nouns are divided into training or testing splits.

**Few Shot Compositional Action Recognition**. As mentioned in Section 3.4 of the main paper, we also evaluate on the few-shot compositional action recognition task in [59]. For this setting, we have 88 *base* action categories and 86 *novel* action categories. We train on the base categories (113K/12K for training/validation) and finetune on few-shot samples from the novel categories (for 5-shot, 430/50K for training/validation; for 10-shot, 860/44K for training/validation).

**Optimization details**. We train using 16 frames with sample rate 4 and batch-size 128 (comprising 64 videos and 64 images) on 8 RTX 3090 GPUs. We train our network for 100 epochs with Adam optimizer [46] with a momentum of $9e-1$ and Gamma $1e-1$. Following [53], we use $lr = 1e-4$ with half-period cosine decay. Additionally, we used Automatic Mixed Precision, which is implemented by PyTorch.

**Regularization details**. We use weight decay of $1e^{-4}$, and a dropout [37] of $5e - 1$ before the final classification.

**Training details**. We use a standard crop size of 224, and we jitter the scales from 256 to 320. Additionally, we set $\lambda_{Con} = 2, \lambda_{HAOG} = 2, \lambda_{Vid} = 1$.

**Inference details**. We take 3 spatial crops per single clip to form predictions over a single video in testing.

### C.4 Something-Something v2

**Dataset**. The SSv2 [59] contains 174 action categories of common human-object interactions.

**Optimization details**. For the standard SSv2 [59] dataset, we train using 16 frames with sample rate 4 and batch-size 128 (comprising 64 videos and 64 images) on 8 RTX 3090 GPUs. We train our network for 100 epochs with Adam optimizer [46] with a momentum of $9e-1$ and Gamma $1e-1$. Following [53], we use $lr = 1e-4$ with half-period cosine decay. Additionally, we used Automatic Mixed Precision, which is implemented by PyTorch.

**Regularization details**. We use weight decay of $1e-4$, and a dropout [37] of $5e-1$ before the final classification.

**Training details**. We use a standard crop size of 224, and we jitter the scales from 256 to 320. Additionally, we use RandAugment [18] with maximum magnitude 9. We set $\lambda_{Con} = 2, \lambda_{HAOG} = 2, \lambda_{Vid} = 1$.

**Inference details**. We take 3 spatial crops per single clip to form predictions over a single video in testing as done in [53].

### C.5 AVA

**Architecture**. SlowFast [22] and MViT-v2 [53] are using a detection architecture with a RoI Align head on top of the spatio-temporal features. We followed their implementation to allow a direct comparison. Next we elaborate on the RoI Align head proposed in SlowFast [22]. First, we extract the feature maps from our SViT model by using the RoIAlign layer. Next, we take the 2D proposal at a frame into a 3D RoI by replicating it along the temporal axis, followed by a temporal global average pooling. Then, we max-pooled the RoI features and fed them to an MLP classifier for prediction.

**Optimization details**. To allow a direct comparison, we used the same configuration as in MViT-v2 [53]. We trained 16 frames with sample rate 4, depth of 16 layers and batch-size 32 on 8 RTX 3090 GPUs. We train our network for 30 epochs with an SGD optimizer. We use $lr = 0.03$ with a weight decay of $1e-8$ and a half-period cosine schedule of learning rate decaying.

**Training details**. We use a standard crop size of 224 and we jitter the scales from 256 to 320. We use the same ground-truth boxes and proposals that overlap with ground-truth boxes by $IoU > 0.9$ as in [22]. We set $\lambda_{Con} = 0.1, \lambda_{HAOG} = 0.5, \lambda_{Vid} = 1$.

**Inference details**. We perform inference on a single clip with 16 frames. For each sample, the evaluation frame is centered in frame 8. We use a crop size of 224 in test time. We take 1 spatial crop with 10 different clips sampled randomly to aggregate predictions over a single video in testing.

## D Additional Datasets Details

Here provide detailed information about the "auxiliary image datasets" (Section D.1) as well as the licenses and privacy policies for these datasets (Section D.2).

### D.1 Auxiliary Image Datasets

In Table 7 we explicitly describe the auxiliary image datasets used in each experiment. As an example, in row 1, we describe the data used for training the SViT-SFT model for the Compositional Action Recognition (CAR) task on the SomethingElse dataset. We note that we began by pretraining on Ego4D and 100DOH.

**Image Annotations**. The collected object boxes from SSv2, 100DOH, Ego4D and AVA are purely human annotated. In SSv2, Ego4D and AVA contact relations between the object and hand are not annotated, so we assign the closest object to the hand for each hand. The contact relations for

| Video Dataset | Task | Model | Pretrain Auxiliary Images | Finetune Auxiliary Images |
|---|---|---|---|---|
| SmthElse | CAR | SViT-SFT | Ego4D, 100DOH | - |
| SmthElse | CAR | SViT-DD | Ego4D, 100DOH | Ego4D, 100DOH |
| SmthElse | CAR | SViT×2% | Ego4D, 100DOH | SmthElse×2% |
| SmthElse | CAR | SViT-ID | Ego4D, 100DOH | SmthElse |
| Ego4D | SCTL | SViT-SFT | Ego4D, 100DOH | - |
| Ego4D | SCTL | SViT-DD | 100DOH | 100DOH |
| Ego4D | SCTL | SViT×2% | Ego4D, 100DOH | Ego4D×2% |
| Ego4D | SCTL | SViT-ID | Ego4D, 100DOH | Ego4D |
| Ego4D | SCC | SViT-SFT | Ego4D, 100DOH | - |
| Ego4D | SCC | SViT-DD | 100DOH | 100DOH |
| Ego4D | SCC | SViT×2% | Ego4D, 100DOH | Ego4D×2% |
| Ego4D | SCC | SViT-ID | Ego4D, 100DOH | Ego4D |
| SSv2 | AR | SViT-SFT | Ego4D, 100DOH | - |
| SSv2 | AR | SViT-DD | Ego4D, 100DOH | Ego4D, 100DOH |
| SSv2 | AR | SViT×2% | Ego4D, 100DOH | SSv2×2% |
| SSv2 | AR | SViT-ID | Ego4D, 100DOH | SSv2 |
| Diving48 | AR | SViT-SFT | Ego4D, 100DOH | - |
| Diving48 | AR | SViT-DD | Ego4D, 100DOH | Ego4D, 100DOH |
| AVA | AD | SViT-SFT | Ego4D, 100DOH | - |
| AVA | AD | SViT-DD | Ego4D, 100DOH | Ego4D, 100DOH |
| AVA | AD | SViT×2% | Ego4D, 100DOH | AVA×2% |
| AVA | AD | SViT-ID | Ego4D, 100DOH | AVA |

Table 7: **Auxiliary Image Datasets** The table describes which image datasets were you used for which SViT setup. CAR refers to Compositional Action Recognition, AR refers to Action Recognition, AD refers to Spatio-temporal Action Detection, SCTL to State Change Temporal Localization, SCC to State Change Classification

100DOH are available in the dataset. In Ego4D, we use the image annotated for the State Change Object Detection task.

**Image/Video annotation naming conventions**. The auxiliary image data we used is originated in video frames. The main difference between video frames and a batch of images is the temporal information. Since we do not use the temporal order of the annotated video frames, and practically use them as images, we refer to them in the paper as "image annotations".

## D.2    Licenses and Privacy

The license, PII, and consent details of each dataset are in the respective papers. In addition, we wish to emphasize that the datasets we use do not contain any harmful or offensive content, as many other papers in the field also use them. Thus, we do not anticipate a specific negative impact, but, as with any Machine Learning method, we recommend to exercise caution.