# OpenReview forum: "Bringing Image Scene Structure to Video via Frame-Clip Consistency of Object Tokens"
_NeurIPS.cc/2022/Conference — NeurIPS 2022 Accept_

### Official Review · Reviewer_LCAU · 2022-07-08

**Rating:** 5
**Confidence:** 5
**Soundness:** 3 good
**Presentation:** 3 good
**Contribution:** 2 fair

**Summary:**

The authors extend prior work on utilizing scene structure information to improve action recognition performance to transformers. In particular, their model is jointly trained to detect hands and objects in images and to classify activities in videos. The images can come from the same or from a different dataset. Differently to a naive multi-task approach they introduce dedicated tokens that are used for object detection, which improves the perfromance somewhat. Additional loss encourages consistency between object tokens (which are only supervised in images) in frames encoded separately and as a part of a clip to make sure that they are not ignored during video inference (a form of domain adaptation). This consistency loss brings further improvements. Overall, minor to moderate improvements are demonstrated on several ego-centric actions recognition datasets (and, for some reason, Diving48).

**Questions:**

Please improve overview of the prior work to included methods that use out-of-domain object detection labels to improve action recognition via object graph reasoning and tone down the novelty claims accordingly.

Please provided an analysis of the method's performance on Diving48 to explain why the proposed approach demonstrated the largest improvements on this dataset despite it not containing any objects.

Please report results on a 3rd person action recognition dataset of your choosing which does require object reasoning to demonstrate to what degree the learned object representations transfer to the 3rd person scenario.

**Limitations:**

It's not clear whether the benefits of the proposed approach are limited to the ego-centric domain.

**Strengths And Weaknesses:**

Strengths:

The paper is well written and is easy to follow.

The idea of utilizing scene structure information for improving action recognition perfromance is sound, though not novel.

The proposed video transformer architecture with object tokens is novel to the best of my knowledge.

Non-negligible improvements over baselines are reported on SomethingElse, Ego4d and Diving48.

A thorough ablation study is provided.


Weaknesses:

The authors ignore prior work which utilizes inference on object graphs to improve action recognition without requiring in-domain object detection label (there is at least A Structured Model For Action Detection by Zhang et al., CVPR'19). There is no conceptual novelty in the proposed approach, only technical novelty of adapting those ideas to the transformer architecture (which is still a valuable contribution).

The authors refer to their approach as multi-modal, but images and videos are not different modalities. These claims should be removed.

The biggest improvement is observed on the Diving48 dataset which does not require any object reasoning. This is counterintuitive and not explained.

---

> ### Author Response · Authors · 2022-08-02
> **Response to Reviewer LCAU - 1/2**
>
> Thank you for your insightful and positive comments regarding our paper. We were able to improve our submission based on your comments. As a result of our responses and draft modifications, we hope that you will consider raising your score.
> \
> \
> **Q1: Please report results on a 3rd person action recognition dataset of your choosing which does require object reasoning to demonstrate to what degree the learned object representations transfer to the 3rd person scenario.
> \
> A1:** Thank you for your suggestion. Based on your comment, we evaluated the accuracy on 3rd-person video datasets, such as Kinetics-400 and AVA. Our results for Kinetics-400 with SViT showed an improvement of +1.5 over MViTv2. As for AVA, we evaluated the mAP and received an improvement of +0.7 over MViTv2. We incorporated these experiments in the revised manuscript in A.4 in Supplementary, See L771-L773.
> \
> \
> Finally, we demonstrate to what extent the learned object representations can be used for auxiliary image tasks. We evaluate the ability of object tokens to be utilized explicitly for the auxiliary task as a simple detector of hands and objects in images. This is accomplished by predicting the detections on SomethingElse based on the learned object tokens. The learned object tokens were compared to the MViTv2 model extended with regression and detection heads (similar to DETR [A]). Our model achieved an mAP of 16.8, while the proposed baseline achieved a similar result with an mAP of 15.5. These results suggest that the object tokens learn meaningful and useful representations. We incorporated these experiments in the revised manuscript in A.2 in Supplementary, See L738-L744.
> \
> \
> [A]  "End-to-End Object Detection with Transformers", ECCV 2020, Cairon et al.
> \
> \
> **Q2: There is no conceptual novelty in the proposed approach, only technical novelty of adapting those ideas to the transformer architecture (which is still a valuable contribution).
> \
> A2:** Our approach demonstrates how utilizing the structure of a small number of images only available during training can improve a video model. In our method, we propose shared object prompts that are used in the image domain and the video domain. In this way, the shared object prompts are utilized to learn the shared structured representation between video and image domains, which enhances the self-attention layers with the structured representation for the main video task. As mentioned by other reviewers, this is a form of “a clever solution to distill information” (`DF7L`) by exploiting the design of the transformer architecture to process video and image domains using the same set of shared weights.
> \
> \
> **Q3: Please improve overview of the prior work to included methods that use out-of-domain object detection labels to improve action recognition via object graph reasoning and tone down the novelty claims accordingly.
> \
> A3:** We will improve the overview of the prior work in the revised manuscript (See L324-L335), along with adding the citation for the paper you mentioned: "A Structured Model For Action Detection” by Zhang et al., CVPR'19. As we briefly mentioned in L319-322, there has been a considerable amount of work on improving action recognition using object graph reasoning [5, 24, 30, 31, 37, 63, 68], and the above study should be included in this list. Indeed these works show the importance of using object representations and their interactions in a video model. Our conceptual and technical contributions focus on the interface between images and video, and modeling these within a single transformer model. We have expanded and edited the related work accordingly.

---

> > ### Author Response · Authors · 2022-08-02
> > **Response to Reviewer LCAU - 2/2**
> >
> > **Q4: Please provided an analysis of the method's performance on Diving48 to explain why the proposed approach demonstrated the largest improvements on this dataset despite it not containing any objects.
> > \
> > A4:** We agree that Diving48 is different from the other datasets since it does not contain any objects or clearly visible hands. We believe that its performance can be explained by two factors: (a) It detects the diving human by using the hand object token. (b) It learns to reason about movement trajectories during training, and thus can model the diving trajectory.
> > \
> > \
> > In support of the first point above, in Figure 3 in the Supplementary, we can see that SViT localizes the human on the Diving48 dataset. To further quantify the contribution of such localization, we performed an experiment removing the "hand" tokens from the HAOGs annotation during training. We observe a resulting degradation of 1.7 in top 1 accuracy. This suggests that training on hand detections helps in the Diving dataset. Additionally, when we removed the “object” tokens we saw a degradation of 0.6. This suggests that hand detections are more important for this dataset. We incorporate these experiments in the revised manuscript in A.2 in Supplementary, See L757-762.  The fact that hand detection helps human-body detection suggests that SViT is capable of distilling structured information from objects of similar but different nature (hands vs humans) with a high degree of transferability.
> > \
> > \
> > As an additional comment, Diving48 is considered a small dataset (15K videos, compared to 70K in Ego4D, 169K in SSv2, or 55K in SomethingElse), and we have already demonstrated that SViT-2% is highly data efficient. Therefore, we believe that the size of the dataset is one of the main reasons why SViT is able to achieve good results on Diving48.
> > \
> > \
> > **Q5: The authors refer to their approach as multi-modal, but images and videos are not different modalities. These claims should be removed.
> > \
> > A5:** Thank you for your suggestion. We revised the manuscript to clarify this point (See L52, L68, L122, L295, L298, L302, L306, L309).
> > \
> > \
> > **Q6: It's not clear whether the benefits of the proposed approach are limited to the ego-centric domain.
> > \
> > A6:** We demonstrated that our approach also improved datasets that were not egocentric. We believe that our method is sufficiently robust such that when the hands and objects are not visible, the performance is not significantly affected, while when they are visible, the performance is boosted.

---

> > > ### Comment · Reviewer_LCAU · 2022-08-03
> > > **R2R**
> > >
> > > I thank the authors fro their detailed response. Unfortunately, some of my main concerns are not addressed, making me inclined to decrease the score. I hope we can resolve those issues during the discussion.
> > >
> > > Firstly, the very valuable additional experiments demonstrate that the model mainly benefits from having a structured representation, not from particular implementation details (e.g. focusing on hands and objects). This is especially clear in the Diving experiments where the video domain has nothing to do with either hands or objects but the improvements are the largest. The authors suggest that this is due to the proposed structured representation being more data efficient. This is indeed in line with claims in the prior work (e.g. Zhang et al., CVPR'19, but also Wu and Kra ̈henbu ̈hl, CVPR'21 is extremely relevant and should be discussed in detail). These results further reinforce my point that this paper is not properly positioned with respect to the prior work. It should be upfront about discussing structured video representations proposed in the past and only then discuss the proposed approach for implementing those ideas with a transformer.
> > >
> > > My previous comments about reducing the novelty claims are also not addressed in the revised manuscript. In particular, in L22-33 the authors do not mention that methods that can utilize image annotations to learn structured video representation already exist (e.g.  Zhang et al., CVPR'19). This work is not proposing anything conceptually new here, but simply adapts these ideas to transformer architectures. The introduction has to be updated accordingly.
> > >
> > > The same goes for the added discussion in L331-335: it omits the most important detail - [77] also uses static image object annotations, not labeled video frames, to build its representation, making it significantly closer to the proposed work than the other references.  This needs to be clearly emphasized and also discussed in the introduction.
> > >
> > > Finally, in the rebuttal the authors claim that "Our conceptual and technical contributions focus on the interface between images and video, and modeling these within a single transformer model.". Modeling images and videos with a single model does not constitute neither a conceptual nor a technical contribution since all existing video architectures essentially treat videos as images with and additional dimension (time), so they can naturally be applied on both videos and images without significant modifications.

---

> > > > ### Author Response · Authors · 2022-08-04
> > > > **Response to Reviewer LCAU**
> > > >
> > > > Thank you for your insightful comments. The draft has been updated in L22-L39 in the introduction and in L339-L349 in the related work section. **This update is meant to reflect the fact that there is extensive prior work on structured representations in computer vision, and also work on utilizing image annotations in videos (e.g., by training detectors and using them for object representations in videos). Our focus is on building a transformer-based video model that has a component for modeling structure, and where this component can use static-image supervision.** We hope it better reflects your viewpoint. Please let us know if there are still issues with the updated revision.
> > > >
> > > > Regarding your final point, our answer "Our conceptual and technical contributions focus on the interface between images and video, and modeling these within a single transformer model," is in reference to structured models. Namely, compared to the structured models in previous works, our approach utilizes a transformer to capture the joint structure.
> > > >
> > > > Last, the title has also been also changed to reflect our focus on video transformers to "Bringing Image Structure to Video **Transformers** via Frame-Clip Consistency of Object Tokens."

---

> > > > > ### Comment · Reviewer_LCAU · 2022-08-08
> > > > > **Re:re**
> > > > >
> > > > > I thank the authors for comprehensively updating the positioning of the paper, which fully addresses my concerns.
> > > > >
> > > > > One small comment: you seem to have forgotten to include references in the current version of the pdf.

---

> > > > > > ### Author Response · Authors · 2022-08-08
> > > > > > **Re:re:re**
> > > > > >
> > > > > > Thanks for your insightful comments. We appreciate your efforts in helping us improve our paper. The references in the current version have also been updated.

---

### Official Review · Reviewer_DF7L · 2022-07-11

**Rating:** 6
**Confidence:** 4
**Soundness:** 4 excellent
**Presentation:** 4 excellent
**Contribution:** 3 good

**Summary:**

This paper proposes the StructureViT (SViT) architecture designed to incorporate structured information about hand-object interaction labels in images to aid in video-related classification tasks. The authors propose a modified video-transformer architecture that, in addition to predicting video-level labels, also predicts the hand-object graph structure for annotated images via additional learned 'object tokens'. To distill information of the hand-object graphs of images to the video domain the authors further propose a Frame-Clip Consistency loss, where the hand-object graphs predicted for a video and its individual frames are forced to be consistent. The authors assess their technique on several different datasets and video-related tasks and show improvements in accuracy over the state-of-the-art (SOTA) across-the-board.

**Questions:**

Q. What procedure/losses did the authors employ for the pre-training task?

Q. One of the main weaknesses of the proposed technique is the requirement of a large number of labeled images (~100K) to distill from. While in Table 1 and Table 3, the authors show the effect of reducing the number of annotated images for the Compositional and Few-Shot Action Recognition tasks, I am curious to know how these effects play out for the experiments presented in Table 2 as well. Does the authors' claim that the number of images can be reduced to 2% (or some similar small number) of the annotated image training images hold in general or is it only true for this task?

Q. As an additional ablation, I would be curious to see the results of an experiment where the authors first learn a vision transformer to perform the HAOG task alone and then fine-tune its backbone network for the video-related task.

------------------
Post-rebuttal:

I thank the reviewers for addressing all my concerns. However, after having considered all the other reviews, I share the concerns of the other reviewers in terms of limited novelty of the current work given the existence of prior non-transformer-based works that incorporate structured information into video-based tasks. Hence, I have lowered my original rating by 1 point.

**Limitations:**

Is the requirement for a large number of annotated images a strict one for the success of the proposed method? If it is, I would like or see the authors acknowledge this limitation more clearly.

**Strengths And Weaknesses:**

Originality: The proposed work is novel in many aspects. It is novel in attempting to exploit the transformer architecture's seamless ability to process multiple domains (video and image in this case) using the same set of shared weights. It is also novel in proposing a clever solution to distill information from the image domain to the video domain via the frame-clip consistency loss. Lastly, it is novel in attempting to injecting information of structured hand-object interaction labels to improve several downstream video-related tasks.

Quality: All methodology and experiments in the paper are technically sound and correct.

Clarity: The material is presented in a clear and organized fashion. Some editorial errors are noted below.
- ln 30: "perfect" should be "perfectly"
- ln 129: $r_i$ should be $r_t$
- ln 130: suggested to write this as $T \times ( H \times W + n)$ for improved clarity
- ln 199: should be  "to treat them"
- ln 305: "share" should be "shared"

Significance: This work presents a significant result towards advancing multi-modality processing and information sharing with transformer architectures. Transformers are quickly becoming the dominant architecture for visual information processing surpassing CNNs. They present the additional advantage of being able to align multi-modality information much more seamlessly versus CNNs. Hence, this work is an interesting exploration that advanced our understanding of the use of transformers for multimodal (image and video) information sharing via transformers. The proposed approach is fairly general and may be applicable with minor modifications to other video and image related tasks as well besides HAOG and the video related tasks considered by the authors.

---

> ### Author Response · Authors · 2022-08-02
> **Response to Reviewer DF7L**
>
> Thank you for your insightful and positive comments regarding our paper. We were able to improve our submission based on your comments. Next, we address your concerns.
>
> **Q1: What procedure/losses did the authors employ for the pre-training task?
> \
> A1:** In the pretraining procedure, we use only $L_{Vid}$ (cross-entropy loss for video classification) and $L_{HAOG}$ (HAOG loss for detecting HAOGs) losses, while we do not use the frame-clip consistency loss ($L_{Con}$). The hyperparameters for the pre-training procedure are identical to those that MViT uses.
> \
> \
> **Q2: One of the main weaknesses of the proposed technique is the requirement of a large number of labeled images (~100K) to distill from. While in Table 1 and Table 3, the authors show the effect of reducing the number of annotated images for the Compositional and Few-Shot Action Recognition tasks, I am curious to know how these effects play out for the experiments presented in Table 2 as well. Does the authors' claim that the number of images can be reduced to 2% (or some similar small number) of the annotated image training images hold in general or is it only true for this task?
> \
> A2:** Thank you very much for your suggestion. Following your comment, we incorporated these results in the revised manuscript in A.2 in Supplementary (See L725-L733 and Table 6c).
> We evaluated SViT-2% on the Ego4d, SSv, and Diving48 datasets. The SViT-2% in Diving48 shows an improvement of +5.8 compared to the MViTv2 baseline (while SViT shows an improvement of +6.7). The SViT-2% in SSv2 shows an improvement of +0.6 compared to the MViTv2 baseline (while SViT shows an improvement of +0.8). The SViT-2% in Ego4D shows an improvement of +1.8 compared to the MViTv2 baseline for object state change classification (while SViT shows an improvement of +2.2) and an improvement of 0.163 compared to the MViTv2 baseline for object state change temporal localization (while SViT shows an improvement of 0.187). We also include these results in the table below.
>
> | Dataset | SViT-2% | SViT | MViTv2|
> | ----------- | ----------- | ----------- | ----------- |
> | Diving48| 78.9 | 79.8 | 73.1|
> | SSv2| 68.7 | 68.9 | 68.1|
> |Ego4D Cls. | 73.4 | 73.8 | 71.6|
> |Ego4D Loc. | 0.539  | 0.515 | 0.702|
>
>
> \
> As can be seen from Table 1, the SViT-2% also appears to be effective and performs similarly on other datasets. Accordingly, we conclude that our claim is generally valid based on these experiments.
> \
> \
> **Q3: As an additional ablation, I would be curious to see the results of an experiment where the authors first learn a vision transformer to perform the HAOG task alone and then fine-tune its backbone network for the video-related task
> \
> A3:** We evaluated this experiment and achieved 62.3 (compared to 63.3 of MViTv2). This indicates that training HAOGs as a form of “distillation,” as we do in SViT, is indeed important.  Following your comment, we incorporated these results in the revised manuscript in A.2 in Supplementary (See L734-L737).
> \
> \
> **Q4: Is the requirement for a large number of annotated images a strict one for the success of the proposed method? If it is, I would like or see the authors acknowledge this limitation more clearly.
> \
> A4:** We demonstrated that our method does not require large annotated images. See A3 for reviewer oj8q.
> \
> The results indicate that using too few annotated images (1 image per 100 videos) may result in degradation, but using a relatively small number of images (1 image per 10 videos) is sufficient to achieve good improvement.
> \
> \
> **Clarity:** The manuscript has been revised to correct the errors you pointed out.

---

> > ### Comment · Reviewer_DF7L · 2022-08-10
> > **Re: Response to Reviewer**
> >
> > I thank the authors for thoroughly answering all my concerns.
> >
> > I am reasonably convinced about the general applicability of their proposed approach to several tasks after their provided additional results.

---

### Official Review · Reviewer_bPRy · 2022-07-11

**Rating:** 4
**Confidence:** 4
**Soundness:** 2 fair
**Presentation:** 2 fair
**Contribution:** 1 poor

**Summary:**

Brief Summary: The paper tackles a few-shot / semi-supervised setup where a small number of training samples about hand-object interaction are provided to drive downstream video performance. The two key ideas are to use hand object information (bounding boxes, and contact edges), and using a frame-consistency loss so that the supervision is propagated to other frames which don't have annotations.

Experiments are carried out on Something-Something, Something-Else, Ego4D, Diving48, and show improvements over compared baselines.

**Questions:**

Q1. (Minor) In section 3.1, the dataset numbering is incorrect. (3) is skipped.

Q2. Why does diving-48 show mvit performing so much worse than slow-fast and time-sformer?



**Limitations:**

I feel that authors should emphasize the limitation of using hand-object graph which limits the proposed methods to a subset of videos where hands are prominent and not occluded. Moreover, it is unclear how the proposed method can be extended to more general action videos such as Kinetics data.

**Strengths And Weaknesses:**

Pros:

1. Interesting finding that using 2% of the in-domain data is comparable to using the entire in-domain data.

2. Multiple datasets are considered for evaluation,


Cons:

1. On naming conventions: I don't think it is fair to say annotated video frames count as image-annotations. They are still video annotations, just that they are sparse.

2. On relations to previous work:

(i) The authors should distinguish their work better compared to previous work. For instance, the idea of using object tags has been previous used in [44], so improvements just based on object tags are not exactly unexpected. Similarly, [Ref1] shows using entity-prompt can improve video-text pretraining.

(ii) Structured representations for videos using semantic roles has been previously investigated, see [Ref2] and [Ref3], but discussion on them is missing.

3. On the task setup:

(i) L71 suggests that in training we have access to video-labels and structured scene annotations. It is not clear how scalable this is, how much preprocessing time and annotation it requires.

(ii) Use of Hand-object graphs seems to be relevant for very specific domains. What if hands are not visible, or they are not interacting with a specific object?

4. On data used: According to L196, video frames are annotated, but no details of how they are obtained are provided. Is it completely automatic / semi-automatic / purely human annotated?

5. On model and experiment:

(i) In L163, the motivation for frame-consistency loss is that image loss may not transfer to video loss. It is unclear if the proposed solution is the best way to approach this. For instance, one could simply perturb the image temporal position embedding to any random index. Alternatively, a object tracking system could interpolate / extrapolate the bounding boxes. Some experiments comparing such additional ways is needed.

(ii) The relative performance improvement in Table 1. are very small, and it is unclear if the differences are actually significant.

(iii) Improvements are not always consistent, for instance, in Table 1, MViT v2 MT outperforms SViT-SFT on Top-1 base. Similarly, and perhaps more surprisingly, SViT-2% outperforms SViT-ID on 10-shot case.

(iv) The authors should provide SViT-2% results for Table 2a,b,c as well.

(v) (Minor) Given that the model works for both images and videos, the authors could show results on human object interaction based datasets such as [Ref4]


[Ref1]: Li, Dongxu, Junnan Li, Hongdong Li, Juan Carlos Niebles, and Steven CH Hoi. "Align and Prompt: Video-and-Language Pre-training with Entity Prompts." In Proceedings of the IEEE/CVF Conference on Computer Vision and Pattern Recognition, pp. 4953-4963. 2022.

[Ref2]: Sadhu, Arka, Tanmay Gupta, Mark Yatskar, Ram Nevatia, and Aniruddha Kembhavi. "Visual semantic role labeling for video understanding." In Proceedings of the IEEE/CVF Conference on Computer Vision and Pattern Recognition, pp. 5589-5600. 2021.

[Ref3]: Chen, Brian, Xudong Lin, Christopher Thomas, Manling Li, Shoya Yoshida, Lovish Chum, Heng Ji, and Shih-Fu Chang. "Joint Multimedia Event Extraction from Video and Article." arXiv preprint arXiv:2109.12776 (2021).

[Ref4]: Chao, Yu-Wei, Yunfan Liu, Xieyang Liu, Huayi Zeng, and Jia Deng. "Learning to detect human-object interactions." In 2018 ieee winter conference on applications of computer vision (wacv), pp. 381-389. IEEE, 2018.

========================
Given the extensive response by the authors, I raise my score by 1 point. In particular, the experiments on Kinetics-data, and additional ablative studies are helpful.

However, I am still confused why, according to the new expts, the proposed method shows improvements in the cases where no hands are visible? This might suggest the real improvements are due to some other hyper-parameter, and not HAOG.

---

> ### Author Response · Authors · 2022-08-02
> **Response to Reviewer bPRy - 1/4**
>
> Thank you for your insightful comments regarding our paper. We were able to improve our submission based on your comments. We hope that our responses below and the draft modifications have addressed all of the comments made in the review. Therefore, we would appreciate it if you would consider updating your score. Next, we address your concerns below.
> \
> \
> **Q1: The authors should distinguish their work better compared to previous work. For instance, the idea of using object tags has been previous used in [44], so improvements just based on object tags are not exactly unexpected. Similarly, [Ref1] shows using entity-prompt can improve video-text pretraining.
> \
> A1:**
> Thank you for your suggestion. Following your comments, the manuscript has been revised in L326-L328 to include these papers (the discussion below will be included in the 10-page camera-ready version). In contrast to [44] and [Ref1], SViT does not require object labels, supervision of textual descriptions, or the use of random region crops during training or inference. Furthermore, both [44] and [Ref 1] focus on vision and language, whereas we focus on the video domain. Last, they utilize external pretrained models that were trained on large datasets, while SViT does not.
> \
> \
> More concretely, [44] uses explicit object representations similar to ours, but unlike ours, [44] uses an external pre-trained detector to initialize the object representation and use them as an input, while our approach learns them only during training. In addition, in [44], object tags (classes) are used, which serve as object labels, while we only use general "object" and "hand" tags (and use them as supervision, not as an input).
> \
> \
> [Ref 1] uses pseudo labels of random region crops from a pretrained prompter module, pretrained on 5.5M video-text pairs and inspired by CLIP, as a source of supervision for video and language training. Compared to our approach, this leverages a much greater amount of data. As mentioned above, this work focuses on vision and language.
> \
> \
> **Q2: Structured representations for videos using semantic roles has been previously investigated, see [Ref2] and [Ref3], but discussion on them is missing.
> \
> A2:**
> Following your comments, the manuscript has been revised in L332-L334 to include these data (the discussion below will be included in the 10-page camera-ready version). Structured representations for videos have been studied in the past (e.g., ActionGenome [37], etc.), and this is not our main contribution, which is to leverage structure from images to video. [Ref2] proposes a new framework for video understanding using visual semantic role labeling. They discussed a specific task and proposed a model designed especially for it. Their proposed task models entities in videos. Their approach differs from ours in that our structured representation models the hand-object interactions within a single scene (within a video clip), whereas [Ref2] models the relation between entities appearing in different clips. [Ref 3] proposes a new task that includes the extraction of events from both video and text. In their approach, they focus on training with pairs of "video+text" data and build on the alignment of text and video in these pairs. In our approach, we do not assume paired data, but instead only a small amount of image-only supervision in addition to the video task. We do not even require any alignment between the images and the video task (i.e., images may come from a variety of sources).
> \
> \
> **Q3: L71 suggests that in training we have access to video-labels and structured scene annotations. It is not clear how scalable this is, how much preprocessing time and annotation it requires.
> \
> A3:**
> A3: As mentioned in L21-L24, several works [5, 24, 30, 31, 37, 63, 68] have shown that “object-centric” models  (ORViT, STRG, STIN, etc.) perform well on action recognition tasks. However, these models require structured annotations of video (as well as video labels), which is clearly very expensive, time-consuming, and not scalable. Our proposed approach uses relatively sparsely labeled images (as SViT-2% experiment in Table 1) in contrast to the above object-centric models. This allows our work to be much more scalable. Furthermore, as noted by reviewer `oj8q`, images are considered "relatively low-cost annotations," which strengthens our motivation to incorporate them into our approach.

---

> > ### Author Response · Authors · 2022-08-02
> > **Response to Reviewer bPRy - 2/4**
> >
> > **Q4: Use of Hand-object graphs seems to be relevant for very specific domains. What if hands are not visible, or they are not interacting with a specific object?
> > \
> > A4:** Following your comment, we incorporated the following experiments in the revised manuscript in A.2 in Supplementary (See L701-L711, Table 6b).
> > \
> > We suggest two experiments to investigate your hypothesis in order to verify the usefulness of our approach. We will first test the model on a filtered test split, which contains only videos without hands. The second step will be to test the model on a filtered split containing only videos without objects. In the experiments, we used SomethingElse, which has dense ground-truth annotations, allowing us to filter frames within the videos. Due to the fact that SomethingElse contains many objects without annotations, there is still data available for training (the numbers are provided below).
> > \
> > \
> > (i) After filtering the videos containing (annotated) hands, we tested our model and the MViTv2 model on the filtered hand split (consisting of 5922 videos). MViTv2 achieved an accuracy of 64.6 while our model achieved 66.5. This implies an improvement of +1.9.
> > \
> > (ii) After filtering the videos containing (annotated) objects in more than 40% of the frames (consisting of 5595 videos), we tested our model and the MViTv2 model on the filtered object split. MViTv2 achieved an accuracy of 64.9 while our model achieved 67.0. This implies an improvement of +2.1.
> > \
> > \
> > We can observe that our model outperforms the baseline even when there are no objects or hands in the videos, demonstrating the robustness of our approach. The total improvement (+1.9 and +2.1) is also a little bit lower than before the filtering (+2.5), which indicates that there has been a slight degradation. Even so, it is still valuable to use the hand-object graphs.
> > \
> > \
> > **Q5: In L163, the motivation for frame-consistency loss is that image loss may not transfer to video loss. It is unclear if the proposed solution is the best way to approach this. For instance, one could simply perturb the image temporal position embedding to any random index. Alternatively, a object tracking system could interpolate / extrapolate the bounding boxes. Some experiments comparing such additional ways are needed.
> > \
> > A5:** Following your suggestion, we include additional baselines in the revised manuscript in A.2 in Supplementary (See L712-L724). Specifically, we did the following: (i) We perturb the image temporal position without consistency loss. We refer to this version as SViT-Perturb (and similarly SViT-Perturb-DD and SViT-Perturb-ID). (ii) We predict the HAOG annotations, which are extrapolated from one random frame of a video. This serves as additional supervision (without the consistency loss) since the HAOG annotations correspond to the video frames (we note that SViT does not require such correspondence since it uses only HAOG annotations from single images). (iii) We predict the HAOG of a random frame in a video, and then duplicate it over the temporal dimension and use it in the same manner as in the consistency loss.
> > \
> > \
> > We find that these three baselines lead to worse performance. (i) We obtained SViT-Perturb-DD with 64.1 (while SViT-DD obtained 65.1), and SViT-Perturb-ID with 65.2 (while SViT-ID with consistency loss got 65.8). (ii) The proposed baseline achieved 65.0 compared to our SViT-ID, which achieved 65.8 (and does not require correspondence). (iii) The proposed baseline achieved 65.1 compared to our SViT-ID, which achieved 65.8. Taken together, this demonstrates the importance of our frame-clip consistency loss.
> > \
> > \
> > **Q6: The authors should provide SViT-2% results for Table 2a,b,c as well.
> > \
> > A6:** Thank you very much for your suggestion.
> > \
> > \
> > We incorporated these results in the revised manuscript in A.2 in Supplementary (See L725-L733, Table 6c). Specifically, we have added the SViT-2% results for the Ego4D, SSV2, and Diving48 datasets. The SViT-2% in Diving48 shows an improvement of +5.8 compared to the MViTv2 baseline (while SViT shows an improvement of +6.7). For SSv2, the SViT-2% experiment results in an improvement of +0.6 compared to the MViTv2 baseline (while SViT shows an improvement of +0.8). For Ego4D, the SViT-2% experiment results in an improvement of +1.8 compared to the MViTv2 baseline (while SViT shows an improvement of +2.2) for object state change classification and an improvement of 0.163 compared to the MViTv2 (while SViT shows an improvement of 0.187) baseline for object state change temporal localization. We also include these results in the table below. It can be seen that the SViT-2% also appears effective and provides similar improvements on other datasets to the results shown in Table 1.
> >
> > | Dataset | SViT-2% | SViT | MViTv2|
> > | ----------- | ----------- | ----------- | ----------- |
> > | Diving48| 78.9 | 79.8 | 73.1|
> > | SSv2| 68.7 | 68.9 | 68.1|
> > |Ego4D Cls. | 73.4 | 73.8 | 71.6|
> > |Ego4D Loc. | 0.539  | 0.515 | 0.702|

---

> > > ### Author Response · Authors · 2022-08-02
> > > **Response to Reviewer bPRy - 3/4**
> > >
> > > **Q7: Moreover, it is unclear how the proposed method can be extended to more general action videos such as Kinetics data.
> > > \
> > > A7:** Following your comment, we evaluated the accuracy on Kinetics-400 with SViT and received an improvement of +1.5% over MViTv2. We incorporated this experiment in the revised manuscript in A.4 in Supplementary, See L771-L773.
> > > \
> > > \
> > > **Q8: The relative performance improvement in Table 1 are very small, and it is unclear if the differences are actually significant.
> > > \
> > > A8:** Action recognition improvements are typically in the range of 1-2%, even for high-impact works such as MViT (MViTv2 outperforms MotionFormer by about 2-3%. See Table 1. This shows that the task is challenging, not that our performance improvement is insignificant. Furthermore, we ran the SViT-ID experiment on SomethingElse (Table 1) ten times with different seeds and calculated a 95% confidence interval of $65.8 \pm 0.44$, while the MViTv2 achieved $63.3 \pm 0.47$, indicating the SViT performance is consistently higher than the reported result of MViTv2. We incorporated these results in the revised manuscript in A.3 in Supplementary (See L765-L768). We will add the variance results to the reported mean performances in Table 1 for the camera ready.
> > > \
> > > \
> > > **Q9: Improvements are not always consistent in Table 1
> > > \
> > > A9:** As explained above, Table 1 shows the results of experiments performed on the SomethingElse dataset. It can be seen that the SVIT models (SViT-DD, SViT-2%, SViT-ID; We exclude SViT-SFT, which is a weaker SViT variant that does not use auxiliary images in fine-tuning) outperform the MViT-MT model (which is the strongest baseline) in all settings. The three SViT models perform similarly, up to statistical noise (note that noise here is on the order of 0.4% as noted in A8, probably due to the small size of the dataset). We do not view this as a problem, but rather as conveying that the main improvement in SViT comes from the use of image supervision, and the particular domain and amount of supervision has a minor effect on performance.
> > > \
> > > \
> > > **Q10: Naming conventions: I don't think it is fair to say annotated video frames count as image-annotations. They are still video annotations, just that they are sparse.
> > > \
> > > A10:** The thinking behind this naming convention is that annotations of individual frames in videos can be referred to as “image annotations” because the temporal order is not used. This is also helpful in terms of presentation, in order to differentiate between video and image supervision. The main difference between video frames and a batch of images is the temporal information. Since we do not use the temporal order of the annotated video frames, we refer to them in the paper as “image annotations”. Following your comment, we incorporate a discussion of this point in the revised manuscript. See L870-L873 in the supplementary.
> > > \
> > > \
> > > **Q11: On data used: According to L196, video frames are annotated, but no details of how they are obtained are provided. Is it completely automatic / semi-automatic / purely human annotated?
> > > \
> > > A11:**  Following your comments, the manuscript has been revised in Section 3.1 to include these details, See L209-L212. The object boxes collected from SSv2, 100DOH, and Ego4D represent the original data, which has been manually annotated (as reported in the original papers [50, 60, 28]). In SSv2 and Ego4D, contact relations between the object and hand are not annotated, so we automatically assign the closest object to a hand for each hand. The contact relations for 100DOH are available in the original publication.
> > > \
> > > \
> > > **Q12: (Minor) Given that the model works for both images and videos, the authors could show results on human-object interaction based datasets such as [Ref4 - hoi]
> > > \
> > > A12:** Thank you for your suggestion.
> > > \
> > > \
> > > ​​We agree with the reviewer about the HOI suggestion. We would like to emphasize that the object prompts in our approach are used in order to leverage structured information from the images into videos. We believe that it is an interesting future direction to use structured information from video to image in our method, and we will leave this to future work.
> > > \
> > > \
> > > Nevertheless, we can evaluate the ability of object tokens to be utilized explicitly for the auxiliary task as a simple detector of hands and objects in images. This is accomplished by predicting the detections on SomethingElse based on the learned object tokens. The learned object tokens were compared to the MViTv2 model extended with regression and detection heads (similar to DETR [A]). Our model achieved an mAP of 16.8, while the proposed baseline achieved a similar result with an mAP of 15.5. These results suggest that the object tokens learn meaningful and useful representations. Following your comments, the manuscript has been revised in A.2 (See L738-L744) to include these results.
> > > \
> > > \
> > > [A]  "End-to-End Object Detection with Transformers", ECCV 2020, Cairon et al.

---

> > > > ### Author Response · Authors · 2022-08-02
> > > > **Response to Reviewer bPRy - 4/4**
> > > >
> > > > **Q13: I feel that authors should emphasize the limitation of using hand-object graph which limits the proposed methods to a subset of videos where hands are prominent and not occluded.
> > > > \
> > > > A13:** It is possible that HAOGs may limit the model to certain domains or scenes. However, as we demonstrate empirically here, our method is robust even for datasets without prominent hands, such as Diving48, AVA, and Kinetics, or even when the hands or objects are occluded (see Q4). As a result of this empirical observation, we believe that our method is robust for a wide range of natural videos. In particular, we believe that our method is sufficiently robust such that when the hands and objects are not visible, the performance is not significantly affected, while when they are visible, the performance is boosted. It should be noted, however, that our approach can be applied to any structural scene type, and therefore, modifying the structure annotation could be an exciting direction to inspire future work.
> > > > \
> > > > \
> > > > **Q14: Why does diving-48 show mvit performing so much worse than slow-fast and timesformer?
> > > > \
> > > > A14:** It is possible that Slow-Fast performs better than MViT because Diving48 is considered a relatively small dataset, and therefore we assume that ConvNets perform better since they have a better inductive bias than Transformers. Also, TimeSformer proposed a data-efficient  spatio-temporal divided attention which could also be more effective when dealing with small datasets than vanilla self-attention.
> > > > \
> > > > \
> > > > **Q15: (Minor) In section 3.1, the dataset numbering is incorrect. (3) is skipped.
> > > > \
> > > > A15:** Thank you. We have revised the manuscript in 3.1 to address this issue.

---

> > > ### Comment · Reviewer_bPRy · 2022-08-08
> > > **Confused by new experimental results**
> > >
> > > I am somewhat confused by the takeaways of Q4, why does the proposed method show improvements even when no hands are present? Isn't that opposite of what is expected?
> > >
> > > It would mean the improvements in all cases are actually coming due to some other reason?
> > >
> > > Overall, I am happy with most new experiments and as such will increase my score by 1 point.

---

> > > > ### Author Response · Authors · 2022-08-09
> > > > **Re:re to Reviewer bPRy**
> > > >
> > > > Thank you for your insightful comments.
> > > >
> > > > In our method, we model objects and hands with object tokens. As objects are very general, the object tokens could be used for a wide range of entities in the scene. For example, in Figure 3 in the Supplementary, we can see that SViT localizes the human on the Diving48 dataset. Similar to Q4, the filtered split on SomethingElse does not include hands, but there are still meaningful objects that can be used to predict specific action labels. For example, in the following link, https://authors98741273.github.io/, we present a few video samples from that split that do not contain the hands, but the objects are still clearly important cues in recognizing the actions.
> > > >
> > > > Additionally, we believe that we still observe improvements as our transformer model mainly benefits from having a structured representation rather than from particular implementation details (e.g., hands and objects). Several previous papers (such as `Zhang et al., CVPR'19` and `Wu and Krahenbuhl, CVPR'21`) suggest that structured representation leads to a more data-efficient model, which may explain why our approach is applicable even when hands or objects cannot be fully observed. Our focus is on building a transformer-based video model that incorporates a component for modeling structure, and this component can be supervised using static images. Following the discussion with reviewer `LCAU`, we revised the paper to emphasize that our model leverages structured representations (see the revised paper introduction `L22-L39` and related work `L339-L349`).
> > > >
> > > > We hope our response above and the draft modifications have addressed all your comments. Please let us know if there are still issues.

---

### Official Review · Reviewer_oj8q · 2022-07-12

**Rating:** 6
**Confidence:** 4
**Soundness:** 3 good
**Presentation:** 4 excellent
**Contribution:** 3 good

**Summary:**

This paper proposes an approach to use annotated images from different domain to help with the training of video-based models. The transformer can take both inputs and there are various objectives involved during training for images and videos, respectively. During inference, the only input is video, and the object prompts are used for label prediction.

**Questions:**

1) I think the motivation of this approach is clear, but in terms of the technical contributions, my understanding is that the joint training of video and image inputs is enabled by ViT to tokenize, and there are other works jointly training images and videos [6, 25, 73, 26, 33, 61]. The authors mentioned that this work "involves explicit interactions between the main video task and the auxiliary image task", so can you further elaborate what is the explicit interaction and how is it not applicable to other approaches like [26, 33, 61]?

2) For experimental results, it appears in Table 3 (a) and (b) that the amount of annotated images is not critical, and neither is the type of HAOG attributes. Does it indicate that the auxiliary task is in fact not functioning as providing additional information, but more as a regularization factor? In other words, it might be worth to systematically demonstrate what are the minimum requirements to make the model work, maybe not the best, but reasonably well (e.g., from ~63 to ~65 in Table 3 examples). This will provide guidance when there's no annotation available but possible annotations can be collected with minimum efforts.

3) Is it possible to use the model in a zero-shot setting and compare with CLIP on action recognition tasks like Kinetics? Also, when comparing with MViTv2 results, is it a fair comparison because additional information/computation has been used in the proposed approach?

**Limitations:**

I think they properly addressed the limitations as mentioned in "We do not anticipate a specific negative impact, but, as with any Machine Learning method, we recommend to exercise caution."

**Strengths And Weaknesses:**

The paper is generally well-written and easy to follow. It is well-motivated to use annotated images due to the relatively low-cost annotations. The experiments demonstrate consistent performance gain over existing method like MViTv2, and the ablations are providing a reasonable break-down of improvements coming from different components.

---

> ### Author Response · Authors · 2022-08-02
> **Response to Reviewer oj8q - 1/2**
>
> Thank you for your insightful comments regarding our paper. We were able to improve our submission based on your comments. Next, we address your concerns.
>
> **Q1: What is the explicit interaction between the main video task and the auxiliary image task, and how is it not applicable to other approaches like [26, 33, 61]?**
> \
> **A1:** In our method, we propose shared object prompts that are used in the image domain and the video domain. In this way, the shared object prompts are utilized for learning the shared structured representation between video and image domains, which enhances the self-attention layers with the structured representation for the main video task. As mentioned by other reviewers, this is a form of “a clever solution to distill information” (`DF7L`) by “exploiting the design of the transformer architecture” (`DF7L`, `LCAU`). Our focus is on leveraging image-level scene structure for video understanding. [26, 61] discuss the relation between different image-level tasks, the information they share, and how they can be utilized for other image-level tasks. [33] describes a multi-task learning approach for reinforcement learning, which is not applicable to the domain of real-life videos. In contrast to these works, our approach aims to demonstrate how we can transfer the structure of a scene from an image to a video.
> \
> \
> **Q2: It appears in Table 3 (a,b) that the amount of annotated images is not critical, and neither is the type of HAOG attributes. Does it indicate that the auxiliary task is not functioning as additional information, but more as a regularization?
> \
> A2:** To examine the regularization effect of our approach, we suggest learning HAOGs without any useful information. Thus, we run an experiment in which the HAOGs are completely random. This means that, for each image, a random HAOG is generated by sampling the boxes and their relationships uniformly. We refer to this experiment as SViT-Random-ID. The SViT-Random-ID result on SomethingElse is 50.6 compared to 63.3 for the SViT-ID baseline. This demonstrates that predicting the actual HAOG attributes is important, but that not many annotations are necessary for the auxiliary task. One explanation may be that learning the shared structured representation between the domains is sample efficient, and does not require many annotated images, which we view as a big advantage of our approach. We incorporated the SViT-Random-ID experiment in the revised manuscript in A.2 in Supplementary (See L745-L753)
> \
> \
> **Q3: It might be worth to systematically demonstrate what are the minimum requirements to make the model work, maybe not the best, but reasonably well.
> \
> A3:**
> We have revised the manuscript in A.2 in Supplementary (See L694-L700 and Table 6a) in response to your comments. We explored the minimum requirements to make our model work. We provide the following experiments: A ratio of 1 image to X videos (where X is 1,10, and 100): 65.6 (1-to-1), 64.8 (1-to-10), and 61.6 (1-to-100). We also include these results in the table below. The results indicate that using too few annotated images (1 image per 100 videos) may result in degradation, but using a relatively small number of images (1 image per 10 videos) is sufficient to achieve good improvement.
> | Images-to-Videos Ratio| top-1|
> | :-----------: | ----------- |
> | 1-1 | 65.6 |
> | 1-10| 64.8 |
> | 1-100 | 61.6 |
>
> \
> \
> **Q4: Is it possible to use the model in a zero-shot setting and compare with CLIP on action recognition tasks like Kinetics?
> \
> A4:**
> Following your comment regarding Kinetics, we evaluated the accuracy in the standard setting (not zero-shot) of Kinetics-400 with SViT and received an improvement of +1.5% over MViTv2 (we incorporated this experiment in the revised manuscript in A.4 in Supplementary, See L771-L773). Regarding zero-shot, we assume you are asking about generalization to new action categories with textual descriptions. CLIP may be used in this direction, but this is an orthogonal direction to our current focus, so it is left for future work.

---

> > ### Author Response · Authors · 2022-08-02
> > **Response to Reviewer oj8q - 2/2**
> >
> > **Q5: When comparing with MViTv2, is it a fair comparison because additional information/computation has been used in the proposed approach?
> > \
> > A5:**
> > In our work, we use additional image data, and this data is not required to be in correspondence with the video dataset. Additionally, our approach requires little additional data (2% of the video frames are sufficient to achieve a reasonable improvement, as shown in Table 3b). Therefore, we believe that the comparison is reasonable.
> > \
> > \
> > In addition, we provide the MViTv2-MT baseline, which utilizes the same additional information as we did. MViTv2-MT is a multitask baseline that consists of the HAOG detection task as well as the video downstream task. Tables 1 and 2 show that SViT outperforms it in all tasks evaluated (+2 on SomethingElse, +2.7 on Ego4d, +0.6 on SSv2, +4.3 on Diving48). Regarding computation, the increase in the computation cost of our approach compared to MViTv2 is relatively small: +0.4% in FLOPs and +5% in parameters (we incorporated this information in the revised manuscript in A.2 in Supplementary in L754-L756).

---

### Author Response · Authors · 2022-08-02
**High-Level Summary**

We thank the reviewers for their insightful comments. Three reviewers support accepting the paper. We are encouraged that they found the proposed approach for incorporating structured information about hand-object interaction in images for video-related tasks to be “novel” (`DF7L`), “well-motivated” (`oj8q`), and "a clever solution to distill information," as well as a "novel attempt to inject information about structured hand-object interactions” (`DF7L`). They observed “consistent improvement gain” (`oj8q`) and “non-negligible/significant” improvements across multiple tasks and datasets (`DF7L`, `LCAU`). They also find that the ablation studies provide "a reasonable breakdown of improvements coming from different components" (`oj8q`) and a "thorough" analysis (`LCAU`). Finally, they find the paper “well written and easy to follow” (`oj8q`, `LCAU`), as well as “clear and organized” (`DF7L`). Next, we address the concerns of each reviewer separately.

---

> ### Comment · Reviewer_LCAU · 2022-08-03
> **Not pretty**
>
> I would like to point out that my review explicitly states that the proposed approach is "is sound, though not novel", and I further elaborate on that as follows: "There is no conceptual novelty in the proposed approach, only technical novelty of adapting those ideas to the transformer architecture (which is still a valuable contribution).". Thus I believe that the authors' summary of my novelty assessment of the work is not entirely accurate :)

---

> > ### Author Response · Authors · 2022-08-03
> > **Response to LCAU**
> >
> > Thank you for bringing this issue to our attention. We based the summary above on the sentence in your review: "The proposed video transformer architecture with object tokens is novel to the best of my knowledge." In order to better reflect your viewpoint, we have updated the summary by removing your reviewer-id from this point.

---

### Meta-Review · Area_Chair_Pmew · 2022-08-23

**Recommendation:** Accept
**Confidence:** Less certain

**Metareview:**

This paper proposes StructureViT (SViT), a network architecture to incorporate structured information from images to aid in video tasks. All four reviewers found several aspects of the paper interesting including the ability to use information from just a few images and be beneficial to video tasks. They noted the thorough experimentation on multiple datasets and also found the paper easy to follow. One of the reviewers had concerns about the positioning of the paper. The authors had multiple discussions with this reviewer and were able to comprehensively update their paper and address most concerns, which was commended by the reviewer. Another reviewer had concerns about comparisons and discussions with regards to previous work. The authors did a good job of addressing most of their concerns. One common concern that emerged from the reviews and discussions was the existence of prior work that incorporates structured information into video tasks, thus reducing the novel contributions of this paper. Having read the paper, reviews and discussions carefully, I think the paper improves upon past work and has sufficient novel contributions that are valuable to readers. I recommend acceptance.

**Award:**

No

---

### Decision · Program_Chairs · 2022-09-14

Accept